# Objective Soups: Multilingual Multi-Task Modeling for Speech Processing

**A F M Saif**[1]   **Lisha Chen**[1,4]   **Xiaodong Cui**[2]   **Songtao Lu**[3]   **Brian Kingsbury**[2]   **Tianyi Chen**[1,5]

[1]Rensselaer Polytechnic Institute   [2]IBM Research   [3]The Chinese University of Hong Kong
[4]University of Rochester   [5]Cornell Tech[*]

[1]`saifa@rpi.edu`,   [1,4]`lisha.chen@rochester.edu`,   [2]`{cuix, bedk}@us.ibm.com`,
[3]`stlu@cse.cuhk.edu.hk`,   [1,5]`chentianyi19@gmail.com`

## Abstract

Training a single model for multilingual, multi-task speech processing (MSP) is severely hampered by conflicting objectives between tasks like speech recognition and translation. While multi-objective optimization (MOO) aims to align gradient updates, its effectiveness diminishes as the number of tasks grows, making it difficult to find a common descent direction. This raises a fundamental question: should highly conflicting objectives be optimized jointly or separated into a hierarchical structure? To address this question, this paper investigates three multi-objective MSP formulations, which we refer to as **objective soup recipes**. These formulations apply multi-objective optimization at different optimization levels to mitigate potential conflicts among all objectives. To ensure efficiency, we introduce a lightweight layer-selection mechanism that computes the conflict-avoiding gradient using only the most problematic layers, minimizing computational and memory overhead. Extensive experiments on CoVoST v2, LibriSpeech, and AISHELL-1 reveal that a bi-level recipe separating recognition and translation tasks consistently outperforms standard flat optimization. Our work demonstrates that hierarchical MOO is a more effective and scalable approach for building state-of-the-art MSP models. Our code has been released at `https://github.com/afmsaif/Objective_Soups`.

## 1 Introduction

Multilingual speech processing is the backbone of voice-driven applications, from virtual assistants to voice search [25, 28]. Modern deployments increasingly demand a *single* model that can (1) transcribe speech in many languages, and (2) perform an auxiliary speech-to-text translation task [55, 8]. Unifying these capabilities simplifies maintenance and reduces inference costs, yet joint training remains difficult due to language diversity, task heterogeneity, and model complexity [30, 21, 46].

A common approach is to introduce one loss term for each task or requirement, for example, a self-supervised learning loss to learn robust representations across languages [41, 48, 3, 4, 29], language-specific Connectionist Temporal Classification (CTC) loss for transcription [34], cross-entropy for translation [60], and fairness constraints for underrepresented languages [10, 11], and then optimize them simultaneously. We refer to this methodology as *objective soups*, where the idea is to leverage multiple objectives to learn a single model that satisfies the above requirements. A related line of work is *rewarded soups* [44], where the goal is to achieve multi-reward alignment for a model by interpolating weights fine-tuned on diverse reward objectives. Although simple and efficient, such

---

[*]This work was conducted when L. Chen and T. Chen were at Rensselaer Polytechnic Institute, and when S. Lu was at IBM Research, United States. The work was supported by IBM through the IBM-Rensselaer Future of Computing Research Collaboration.

soups could suffer from *gradient conflicts*: improving one objective can actively harm the others, stalling overall training. *Multi-objective optimization* provides a principled approach for handling conflicting training objectives [20, 37, 36, 9, 19]. We propose a hybrid multi-level and multi-objective optimization algorithm tailored to MSP: in our formulation, the self-supervised loss serves as a lower-level constraint that drives the model toward language-agnostic acoustic representations. We first optimize this lower level, then feed its solution into the upper-level supervised objective—combining CTC for transcription and cross-entropy for translation—by adding a penalty term. Allowing the representation-oriented self-supervised objective to converge first yields a more stable optimization path than a flat objective soup, and this bilevel structure naturally extends to additional hierarchies (e.g., grouping objectives by tasks or languages) [49].

**Our findings and contributions.** We propose three optimization recipes: i) VS-MSP: a single-level vector-optimization approach; ii) VC-MSP: a *bilevel* approach that places self-supervised learning in the lower level and supervised fine-tuning in the upper level; and, iii) VM-MSP: a *multilevel* approach that organizes objectives hierarchically by tasks or by languages, so that the most conflicting objectives are separated across levels. Experiments on CoVoST v2, LibriSpeech, and AISHELL datasets, covering model sizes from 58M to 150M parameters, reveal the following patterns:

**F1:** **Multi-objective optimization reduces gradient conflicts and boosts performance.** VC-MSP outperforms the classic pre-training and fine-tuning baseline by 9.8% in terms of WER and 8.6% in terms of BLEU.

**F2:** **Introducing hierarchies across objectives further improves accuracy.** The multi-level approach VM-MSP outperforms VC-MSP, delivering 4.2% WER and 2.8% BLEU gains.

**F3:** **Task-based hierarchies outperform language-based ones.** Separating speech recognition and translation at different levels yields larger gains than separating by language, yielding up to 7.4% in terms of WER and 8.5% in terms of BLEU improvements over baselines.

**F4:** **Layer-wise conflict pruning reduces overhead without losing accuracy.** We select layers whose gradients have negative cosine similarity, and compute the conflict-avoidant dynamic update direction[2] only on this subset of gradients. By pruning layers with aligned gradients, we reduce up to 17% in time and 18% in memory, without degrading WER or BLEU.

From these findings, we conclude that multiobjective optimization, with a carefully defined optimization hierarchy and a lightweight layer-selection mechanism, offers an effective and practical recipe for multilingual multi-task speech recognition and translation.

## 2 Related Work

In this section, we briefly review related work on multi-objective optimization, multilingual speech recognition, and translation; see also Appendix A for additional related work.

**Multi-objective optimization.** Classical multi-objective optimization methods focus on optimizing multiple conflicting objectives, with approaches like scalarization and lexicographic ordering [39, 38, 51, 36, 9, 19]. In speech processing, multi-objective optimization has been applied at the system level to balance accuracy and model size [40]. In contrast, our work addresses objective conflict at the training level, optimizing multiple loss functions jointly within a shared multilingual model.

**Multilingual speech recognition and translation.** Earlier approaches in multilingual speech recognition relied on deep neural networks, hidden Markov models, and LSTM models, gradually progressing to Seq2Seq and transformer-based architectures [27, 24, 58, 6]. While significant advancements have been made, challenges remain in enabling multi-tasking and reducing conflicts across objectives-a gap that this paper addresses.

**Multi-task learning for speech.** Multi-task learning for speech recognition and translation tasks has seen limited exploration but includes methods such as dual encoder-decoder architectures [33] and large-scale multitask models like Whisper [43]. Recent work, such as Mu$^2$SLAM [13], incorporates cross-modality learning, while others use joint pre-training and fine-tuning [5, 47]. However, most methods rely on static weighting strategies, which do not adequately address conflicting objectives.

This paper investigates conflicting objectives in MSP and proposes three algorithms to mitigate these conflicts. Our approach demonstrates a significant improvement over baseline methods, highlighting the effectiveness of multi-objective optimization in multilingual MSP tasks.

---

[2]Definition of the conflict-avoidant dynamic update direction is provided in Section 3

# 3 Unifying Multi-Objective Optimization Training Methods

In this section, we introduce multi-objective optimization, its optimality condition, discuss three problem formulations, and present the corresponding algorithms to solve these problems.

## 3.1 Multi-objective optimization: a primer

Multi-objective optimization aims to learn a model that simultaneously optimizes multiple conflicting objectives, which can represent different tasks or learning metrics [36, 9]. Let $\Theta \in \mathbb{R}^q$ denote the model parameter. Given $M$ objectives with each denoted as $\ell_m(\Theta)$, for $m \in [M]$, the general multi-objective optimization problem is to solve

$$\min_{\Theta \in \mathbb{R}^q} \mathcal{L}(\Theta) := [\ell_1(\Theta), \ldots, \ell_M(\Theta)]. \tag{1}$$

Since optimizing multiple objectives simultaneously is often challenging, understanding the notion of *conflict* is crucial, which is defined below.

**Definition 3.1** (Gradient conflict). For any pair $i, j \in [M]$, let $\ell_i(\Theta)$ and $\ell_j(\Theta)$ be the loss functions for two different languages or MSP tasks, parameterized by $\Theta$. We say a gradient conflict exists if $\cos(\nabla_\Theta \ell_i, \nabla_\Theta \ell_j) < 0$, where $\cos(\cdot)$ is the cosine similarity function.

The definition of gradient conflict indicates that improving one objective along its gradient degrades the other, necessitating trade-offs for balanced optimization. Next we introduce the necessary optimality condition for multi-objective optimization.

**Definition 3.2** (Pareto stationary). A model $\Theta$ is Pareto stationary if there exists $\lambda \in \Delta^M := \{\lambda \in \mathbb{R}^M \mid \mathbf{1}^\top \lambda = 1, \ \lambda \geq 0\}$ such that $\nabla \mathcal{L}(\Theta)\lambda = 0$, which is equivalent to $\min_{\lambda \in \Delta^M} \|\nabla \mathcal{L}(\Theta)\lambda\| = 0$.

We denote the model parameters by $\Theta := \{\theta, \phi\}$, where $\theta$ is the parameter of the backbone and $\phi$ is for a language/task-dependent layer; see an illustration in Figure 2. Before formulating the multi-objective problems, we specify the objectives.

**Objectives of self-supervised and supervised training.** For pre-training shared backbone parameters $\theta$, we employ the Contrastive Predictive Coding (CPC) loss $\ell_u(\theta)$ [41] for self-supervised training to learn good

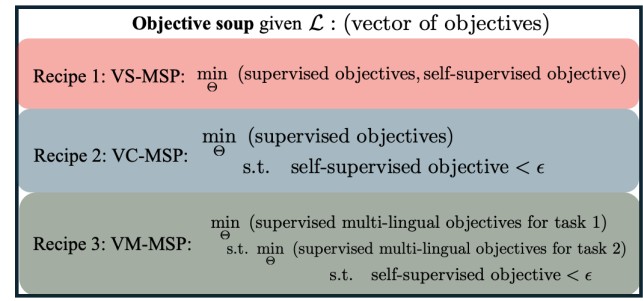

Figure 1: An overview of the objective soup and the three recipes.

language-agnostic acoustic features. For language and task specific parameters $\phi_{t,n}$, we use supervised loss, $\ell_s(\theta, \phi_{t,n})$, where $t \in [T]$ and $n \in [N]$ represent different languages and tasks, respectively. This can be either the CTC loss $\ell_{ctc}(\theta, \phi)$ [23] for transcription and the cross-entropy loss $\ell_{ce}(\theta, \phi)$ for translation.

A set of self-supervised losses and multiple supervised losses form the objectives for multilingual MSP. The use of self-supervised loss during the supervised training restricts the search region to a neighborhood of models with good representation capability that continues to intersect the Pareto set [47, 16]. This neighborhood refers to the sublevel set $\mathcal{R}_\delta := \{\theta : \ell_u(\theta) \leq \ell_u^\star + \delta\}$ of the self-supervised

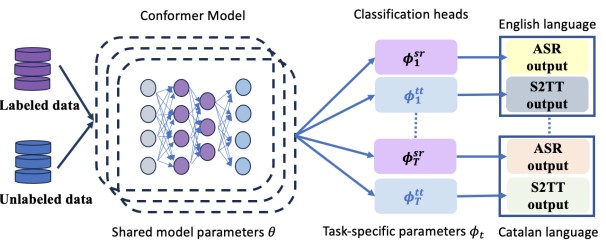

Figure 2: Multi-head conformer model.

loss around its minimum $\ell_u^\star := \min_\theta \ell_u(\theta)$, which keeps the encoder's invariances intact. Operationally, we do not project onto $\mathcal{R}_\delta$. Instead, our update uses a penalty term $\eta \left( \ell_u(\theta) - \ell_u^\star \right)$ to enforce the constraint and bias the iterates to remain near $\mathcal{R}_\delta$. This corresponds to the $\varepsilon$-constraint viewpoint in multi-objective optimization [39, §3.2], where increasing $\eta$ approximates restricting the feasible set to $\mathcal{R}_\delta$. We then jointly optimize the self-supervised and supervised multi-lingual multi-task losses

using the proposed *objective soup* approach. Specifically, we propose three different recipes that use a different hierarchical modeling of the objectives, placing different objectives at different optimization levels. See Figure 1 and a detailed discussion thereafter.

Note that for MSP, we can represent all the objectives as a vector $\mathcal{L}(\Theta)$ containing both self-supervised loss and supervised losses from different languages and tasks such as multilingual speech recognition and translation, where $\Theta := \{\theta, \phi_{1,1}, \cdots, \phi_{T,N}\}$; that is, $\mathcal{L}(\Theta) := [\ell_s(\theta, \phi_{1,1}), \ldots, \ell_s(\theta, \phi_{T,N})]$.

To encourage Pareto stationarity for objectives of different MSP variants, we can employ either static or dynamic weighting multi-objective optimization methods.

**Limitation of static weighting.** In static weighting, we optimize the (weighted) average of the multiple objectives [32, 59]. This method is simple but may suffer from conflicting objectives where gradients have conflicting directions. For instance, considering $\ell_{t,n}(\Theta) = \ell_s(\theta, \phi_{t,n}) + \eta\ell_u(\theta)$ and $\ell_{t',n'}(\Theta) = \ell_s(\theta, \phi_{t',n'}) + \eta\ell_u(\theta)$ two objectives having conflicting directions, $(t, t') \in [T]$ and $(n, n') \in [N]$, then $\langle \nabla_\Theta \ell_{t,n}(\Theta), \nabla_\Theta \ell_{t',n'}(\Theta) \rangle < 0$.

**Proposed dynamic weighting.** To avoid conflicting directions, we can employ a dynamic weighting method that uses dynamically weighted gradients from individual objectives to avoid conflict and enables optimization in a conflict-avoiding (CA) direction [9]. Specifically, a CA direction $d$ is the steepest common descent direction that maximizes the worst descent, given by

$$d(\Theta) = \arg\max_d \min_{\lambda \in \Delta^{NT}} -\langle \nabla\mathcal{L}(\Theta)\lambda, d \rangle - \frac{1}{2}||d||^2. \tag{2}$$

By reformulation, such a direction is equal to dynamically weighted gradients of different objectives [9], given by $d(\Theta) = -\nabla\mathcal{L}(\Theta)\lambda^*(\Theta)$ with weight $\lambda^*(\Theta)$ computed by

$$\lambda^*(\Theta) = \arg\min_{\lambda \in \Delta^{NT}} \|\nabla\mathcal{L}(\Theta)\lambda\|^2. \tag{3}$$

However, finding the true gradients of $\nabla\mathcal{L}(\Theta)$ is computationally expensive. Hence, we employ a stochastic variant of MGDA, the MoDo algorithm [9], which obtains an unbiased stochastic gradient estimate for (3) via a double sampling technique.

At each iteration $k$, denote $\xi_1^k$ and $\xi_2^k$ as two independent samples from the labeled dataset $D$, and $\nabla\ell(\xi_1^k; \Theta^k)$ and $\nabla\ell(\xi_2^k; \Theta^k)$ as the stochastic gradients. We leverage the MoDo update in [9] by

$$\lambda^{k+1} = \Pi_{\Delta^{NT}}\left(\lambda^k - \gamma^k\left(\nabla\mathcal{L}(\xi_1^k; \Theta^k)^\top \nabla\mathcal{L}(\xi_2^k; \Theta^k)\right)\lambda^k\right) \tag{4}$$

where $\gamma^k$ is the step size, $\Pi_{\Delta^{NT}}(\cdot)$ denotes the projection to $\Delta^{NT}$; see a summary in Table 1.

Having introduced the basics of multi-objective optimization, we formulate the three multi-objective MSP problems and define their corresponding parameter update rules. For brevity, we consider only the three objectives mentioned above; the generalized formulation is provided in the Appendix B.

### 3.2 Vectorized single-level MSP (VS-MSP)

In this formulation, we treat all objectives as a single-level vectorized objective without any lower-level constraints. Hence, the problem formulation is:

$$\min_{\Theta \in \mathbb{R}^q} \mathcal{L}(\Theta) := \Big[\underbrace{\ell_{ctc}(\theta, \phi_{1,1}), \ell_{ce}(\theta, \phi_{1,2})}_{\text{1-st language with 2 tasks}}, \ldots, \underbrace{\ell_{ctc}(\theta, \phi_{T,1}), \ell_{ce}(\theta, \phi_{T,2})}_{T\text{-th language with 2 tasks}}, \ell_u(\theta)\Big]. \tag{5}$$

For single vectorized objective training, we can optimize the vectorized objectives using Algorithm 1 in Appendix B where the shared backbone parameters are updated using

$$\theta^{k+1} = \theta^k - \alpha \sum_{t=1}^T \lambda_{t,ctc}^k \nabla_\theta \ell_{ctc}(\theta^k, \phi_{t,1}^k) - \alpha \sum_{t=1}^T \lambda_{t,ce}^k \nabla_\theta \ell_{ce}(\theta^k, \phi_{t,2}^k) - \alpha\lambda_u^k \nabla_\theta \ell_u(\theta^k) \tag{6}$$

where $\alpha > 0$ is the learning rate for the backbone parameters, and $\lambda_{t,ctc}^k$ and $\lambda_{t,ce}^k$, which represent the dynamic weights for speech recognition and translation objectives, are computed using the MoDo algorithm [9]. Here, $\lambda_u^k$ is the dynamic update direction for the self-supervised objective, $\ell_u$, with $\sum_{t=1}^T \lambda_{t,ctc}^k + \sum_{t=1}^T \lambda_{t,ce}^k + \lambda_u^k = 1$. Similarly, by taking the gradients of each supervised objective with respect to task-specific output heads, the task-specific parameters are updated via

$$\phi_{t,1}^{k+1} = \phi_{t,1}^k - \beta\nabla_\phi \ell_{ctc}(\phi_{t,1}^k, \theta^k) \quad \text{and} \quad \phi_{t,2}^{k+1} = \phi_{t,2}^k - \beta\nabla_\phi \ell_{ce}(\phi_{t,2}^k, \theta^k) \tag{7}$$

where $\beta > 0$ is the learning rate of the task-specific parameter.

## 3.3 Vectorized objectives with lower-level constraint for MSP (VC-MSP)

To mitigate the challenge of conflicting objectives and reduce the search space for an optimal Pareto stationary point, incorporating a suitable lower-level constraint, $\ell_{\mathrm{u}}(\theta)$, can be beneficial [39]. However, $\ell_{\mathrm{u}}(\theta)$ must satisfy an essential property: its gradient update direction should exhibit minimal conflict with the upper-level objectives. When such alignment holds, the constraint restricts the feasible region to a representation-preserving neighborhood that still intersects the Pareto set and admits a common descent direction for gradient-based solvers. If the lower-level objectives are strongly conflicting, the constraint does not aid optimization. In this context, we incorporate the self-supervised loss as a lower-level constraint, as it exhibits this desirable property (see Appendix E). This approach helps align the gradient directions and maintain a feasible optimization region, ultimately enhancing overall performance. By constraining the self-supervised loss to be smaller than a threshold $\varepsilon$, our VC-MSP method can be formulated as

$$\min_{\Theta \in \mathbb{R}^q} \quad \mathcal{L}(\Theta) := \Big[ \underbrace{\ell_{\mathrm{ctc}}(\theta, \phi_{1,1}), \ell_{\mathrm{ce}}(\theta, \phi_{1,2})}_{\text{1-st language with 2 tasks}}, \ldots, \underbrace{\ell_{\mathrm{ctc}}(\theta, \phi_{T,1}), \ell_{\mathrm{ce}}(\theta, \phi_{T,2})}_{T\text{-th language with 2 tasks}} \Big] \quad \text{(8a)}$$

$$\text{s.t.} \quad \ell_{\mathrm{u}}(\theta) - \min_{\theta'} \ell_{\mathrm{u}}(\theta') \le \varepsilon. \quad \text{(8b)}$$

This formulation minimizes the vector of supervised losses $\mathcal{L}(\Theta)$ subject to the constraint $\ell_{\mathrm{u}}(\theta) - \ell_{\mathrm{u}}^\star \le \varepsilon$, where $\ell_{\mathrm{u}}^\star := \min_\theta \ell_{\mathrm{u}}(\theta)$. The $\varepsilon$–constraint defines the feasible region for the supervised objectives. We optimize this constrained problem via its penalized first–order realization (see Eq. (15)), which *targets Pareto–stationary solutions* for the constrained setting—i.e., limit points at which no feasible common descent direction exists that decreases all supervised objectives simultaneously [39]. Empirically, placing the self–supervised objective at the lower level and ASR/translation at the upper level yields the best validation performance in our setting. This observation motivates the VC–MSP algorithm: we separate the self–supervised objective from the supervised ones and optimize them at the lower and upper levels, respectively, using a single training loop with a penalty schedule.

To train a model using the VC-MSP algorithm, the backbone $\theta$ is updated via

$$\theta^{k+1} = \theta^k - \alpha \sum_{t=1}^{T} \lambda_{t,\mathrm{ctc}}^k \nabla_\theta \ell_{\mathrm{ctc}}(\theta^k, \phi_{t,1}^k) - \alpha \sum_{t=1}^{T} \lambda_{t,\mathrm{ce}}^k \nabla_\theta \ell_{\mathrm{ce}}(\theta^k, \phi_{t,2}^k) - \alpha\eta \nabla_\theta \ell_{\mathrm{u}}(\theta^k) \quad \text{(9)}$$

where $\sum_{t=1}^{T} \lambda_{t,\mathrm{ctc}}^k + \sum_{t=1}^{T} \lambda_{t,\mathrm{ce}}^k = 1$. To update task-specific heads, we use (7); see Algorithm 2.

## 3.4 Vectorized multilevel MSP (VM-MSP)

Building upon the VC-MSP formulation, we introduce VM-MSP, a multilevel MSP algorithm. With VM-MSP, we aim to explore whether extending our VC-MSP algorithm into a multi-level optimization based on tasks and languages offers advantages and mitigates the risk of being trapped in sub-optimal Pareto stationary points. In multi-level optimization, it follows a hierarchical structure, with

| Method | Backbone update rule |
|---|---|
| VS-MSP | $\theta^{k+1} = \theta^k - \alpha \left[ \sum_{t,n} \lambda_{t,n}^{(k)} \nabla_\theta \ell_n \;+\; \lambda_u^{(k)} \nabla_\theta \ell_u \right]$ |
| VC-MSP | $\theta^{k+1} = \theta^k - \alpha \left[ \sum_{t,n} \lambda_{t,n}^{(k)} \nabla_\theta \ell_n \;+\; \eta \nabla_\theta \ell_u \right]$ |
| VM-MSP | $\theta^{k+1} = \theta^k - \alpha \left[ \sum_{t,n \in \mathcal{T}_1} \lambda_{t,n}^{(k)} \nabla_\theta \ell_n \;+\; \eta_1 \sum_{t,n \in \mathcal{T}_2} \lambda_{t,n}^{(k)} \nabla_\theta \ell_n \;+\; \eta \nabla_\theta \ell_u \right]$ |

Table 1: The red boxes represent updates for the objectives with gradient conflict mitigation, the blue box represents the update for the supervised objective as the penalty terms, the teal boxes represent updates for the self-supervised objective as a penalty term.

decisions made at different levels within the hierarchy. The problem formulation for multilevel MSP optimization can be expressed as follows:

$$\min_{\Phi_1 \in \mathbb{R}^T, \Phi_2^*, \theta} \mathcal{L}_{\mathrm{ctc}}(\Phi_1, \Phi_2^*, \theta)$$

$$\text{s.t.} \quad \Phi_2^* = \operatorname*{argmin}_{\Phi_2 \in \mathbb{R}^T, \theta} \mathcal{L}_{\mathrm{ce}}(\Phi_1, \Phi_2, \theta) \quad \text{(10)}$$

$$\text{s.t.} \quad \ell_{\mathrm{u}}(\theta) - \min_{\theta'} \ell_{\mathrm{u}}(\theta') \le \varepsilon$$

where $\Phi_1 := \{\phi_{1,1}, \ldots, \phi_{T,1}\}$ and $\Phi_2 := \{\phi_{1,2}, \ldots, \phi_{T,2}\}$.

In VM-MSP, training is performed at multiple levels with feedback across different levels. Specifically, we update the backbone parameters $\theta$ via

$$\theta^{k+1} = \theta^k - \alpha \left( \sum_{t=1}^{T} \lambda_{t,\text{ctc}}^k \nabla_\theta \ell_{\text{ctc}}(\theta^k, \phi_{t,\text{ctc}}^k) + \eta_1 \sum_{t=1}^{T} \lambda_{t,\text{ce}}^k \nabla_\theta \ell_{\text{ce}}(\theta^k, \phi_{t,\text{ce}}^k) + \eta \nabla_\theta \ell_{\text{u}}(\theta^k) \right) \quad (11)$$

where $\eta_1$ and $\eta$ are penalty parameters, with $\eta_1$ controlling the relative influence of the translation loss with respect to the ASR loss, and $\eta$ controlling the influence of the self-supervised loss as a lower-level constraint in the multilevel optimization. Here, $\sum_{t=1}^{T} \lambda_{t,\text{ctc}} = 1$ and $\sum_{t=1}^{T} \lambda_{t,\text{ce}} = 1$. We update task-specific classification parameters using (7); see a summary in Algorithm 3 of Appendix F. Note that in this formulation, the unsupervised loss is optimized first, followed by the translation objective, and then the ASR objective. We can also change the optimization order; the unsupervised objective is optimized first, followed by the ASR objective, and then translation.

*Remark* 3.3. For multilevel optimization, objectives are prioritized based on their importance. In VM-MSP, this includes task-based and language-based multilevel optimization. Task-based multilevel optimization experiments with speech recognition and translation, alternating their primary and secondary levels. Language-based multilevel optimization involves English (LibriSpeech) and Chinese (AISHELL), alternating their primary and secondary levels.

To update the backbone parameters, $\theta$, and task-specific parameters, $\phi$, in the three algorithms, we use first-order gradient-based updates. A summary is provided in Table 1 with detailed descriptions and derivations deferred to Appendix B.

## 4 Experimental Results and Findings

In this section, we compare our three proposed MSP algorithms with several baselines: two-stage training (pre-training then fine-tuning), static weighting (fine-tuning with tuned loss weights), and joint (bilevel) training. Our goal is to identify the most effective method for resolving conflicting objectives, thereby avoiding the risk of the model getting stuck in a suboptimal Pareto stationary point. We analyze speech recognition and translation performance in a multilingual setup using the CoVoST 2 dataset, selecting five languages for speech recognition (English (En), French (Fr), German (De), Spanish (Es), Catalan (Ca)) and four for translation (Fr, De, Es, Ca). Additionally, we conduct experiments with a combination of the LibriSpeech and AISHELL datasets. Our results consistently demonstrate that our approaches outperform the baselines, confirming their effectiveness in achieving superior speech recognition and translation performance.

**Models and hyper-parameters:** In our experiments, we evaluate three encoder–decoder architectures: Conformer + Transformer decoder. Conformer [26] A uses 12 Conformer blocks with a model dimension of $d = 612$ and $H = 12$ attention heads (head size 51); Conformer B uses 8 Conformer blocks with $d = 512$ and $H = 8$ heads (head size 64). Both employ a convolutional kernel of size 31. Each model includes a speech recognition classification head consisting of a dropout layer (rate $\delta = 0.1$) followed by a linear projection to the speech recognition vocabulary size $V_{\text{ASR}}$, producing logits that are trained with the CTC loss. The translation component is a standard Transformer decoder with $L_e = 3$ encoder layers and $L_d = 3$ decoder layers, model dimension $d$ matching the Conformer encoder (612 for A, 512 for B), $H = 12$ *or* 8 attention heads, feed-forward dimension $d_{\text{ff}} = 2048$, dropout rate $\delta = 0.1$, and learned token embeddings of size $d$. The output logits over the translation vocabulary $V_{\text{ST}}$ are trained with cross-entropy loss.

Whisper-medium. We adopt the medium-sized Whisper model [43] (24 encoder layers, 12 decoder layers, $d = 1024$, $H = 16$, and a feed-forward dimension of 4096). Its encoder processes input audio into hidden representations, while its decoder performs autoregressive decoding for both speech recognition and translation tasks, trained with cross-entropy loss over the output vocabulary.

All models are trained with the AdamW optimizer, using a backbone learning rate of $\alpha = 5 \times 10^{-5}$ and a head/decoder learning rate of $\beta = 5 \times 10^{-4}$. For VC-MSP, the bilevel penalty is initialized to $\eta = 0$ and increased by 0.02 each epoch. For VM-MSP, we set $\eta_1 = 0.1$ and $\eta_2 = 0$, with each being incremented by 0.02 per epoch.

**Training time and memory complexity:** Table 5 reports the GPU memory footprint and per-epoch training time for the baseline two-stage pipeline and our MSP variants. Introducing dynamic

Table 2: Speech recognition and translation on CoVoST-2 using Conformer models comparing two-stage, two-stage static, joint training, VS-MSP, VC-MSP, and VM-MSP.

| Task Param | Lang | Two-stage training | Two-stage static | Joint training | VS-MSP | VC-MSP | VM-MSP UAS | VM-MSP USA | VM-MSP (Efficient) |
|---|---|---|---|---|---|---|---|---|---|
| Speech recognition (WER ↓) (150 M) | En | 22.2% | 22.2% | 21.3% | 21.1% | 20.6% | **19.7%** | 19.9% | 19.6% |
| | Fr | 23.8% | 23.6% | 22.9% | 23.1% | 22.6% | **21.8%** | 22.1% | 21.7% |
| | De | 16.7% | 16.6% | 16.1% | 16.0% | 15.3% | **14.6%** | 14.9% | 14.7% |
| | Es | 18.8% | 18.9% | 18.2% | 18.4% | 17.7% | **17.0%** | 17.2% | 16.9% |
| | Ca | 21.2% | 21.0% | 20.5% | 20.3% | 19.8% | **19.1%** | 19.3% | 19.3% |
| | Ave. | 20.5% | 20.4% | 19.8% | 19.7% | 19.2% | **18.4%** | 18.7% | 18.4 % |
| Speech recognition (WER ↓) (83 M) | En | 28.6% | 28.4% | 27.8% | 27.9% | 27.3% | **26.9%** | 27.2% | 26.8% |
| | Fr | 26.1% | 26.0% | 25.3% | 25.1% | 24.5% | **24.1%** | 24.3% | 24.3% |
| | De | 22.5% | 22.6% | 22.0% | 22.2% | 21.6% | **21.2%** | 21.5% | 20.9% |
| | Es | 23.9% | 23.8% | 23.1% | 23.3% | 22.6% | **22.1%** | 22.3% | 22.2% |
| | Ca | 25.2% | 25.0% | 24.6% | 24.8% | 24.2% | **23.8%** | 24.1% | 23.9% |
| | Ave. | 25.3% | 25.1% | 24.5% | 24.6% | 24.0% | **23.6%** | 23.9% | 23.6 % |
| Translation (BLEU ↑) (150 M) | Fr→En | 27.2 | 27.4 | 28.1 | 28.3 | 28.9 | 29.5 | **29.6** | 29.4 |
| | De→En | 26.9 | 27.1 | 27.9 | 27.7 | 28.2 | 28.5 | **28.7** | 28.5 |
| | Es→En | 29.1 | 29.3 | 30.2 | 30.0 | 30.9 | 31.3 | **31.6** | 31.5 |
| | Ca→En | 22.7 | 22.6 | 23.8 | 23.6 | 24.5 | 25.4 | **25.8** | 25.6 |
| | Ave. | 26.5 | 26.6 | 27.5 | 27.4 | 28.1 | 28.7 | **28.9** | 28.8 |
| Translation (BLEU ↑) (83 M) | Fr→En | 24.2 | 23.5 | 24.1 | 23.9 | 25.8 | 26.5 | **26.6** | 26.4 |
| | De→En | 22.4 | 22.6 | 23.3 | 23.0 | 23.9 | 24.5 | **24.7** | 24.4 |
| | Es→En | 24.6 | 24.5 | 25.1 | 24.8 | 25.6 | 26.1 | **26.5** | 26.8 |
| | Ca→En | 19.7 | 19.5 | 20.2 | 20.5 | 20.8 | 21.3 | **21.6** | 21.4 |
| | Ave. | 22.7 | 22.5 | 23.2 | 23.1 | 24.0 | 24.6 | **24.9** | 24.8 |

Table 3: Speech recognition and translation on CoVoST-2 and Whisper-medium model comparing two-stage, two-stage static, joint training, VS-MSP, VC-MSP, and VM-MSP.

| Task | Lang / Lang→Eng | Two-stage training | Two-stage static | Joint training | VS-MSP | VC-MSP | VM-MSP UAS | VM-MSP USA | VM-MSP (Efficient) |
|---|---|---|---|---|---|---|---|---|---|
| Speech recognition (WER↓) | En | 15.9% | 15.8% | 15.2% | 15.6% | 14.9% | **14.3%** | 14.6% | 14.3% |
| | Fr | 21.7% | 21.7% | 21.1% | 21.5% | 20.6% | **20.1%** | 20.4% | 20.2% |
| | De | 10.4% | 10.2% | 9.3% | 10.1% | 8.9% | **8.2%** | 8.5% | 8.0% |
| | Es | 14.1% | 14.2% | 13.5% | 13.9% | 12.8% | **12.3%** | 12.4% | 12.2% |
| | Ca | 17.5% | 17.3% | 16.7% | 16.9% | 16.6% | **15.9%** | 16.2% | 16.0% |
| | Ave. | 15.9% | 15.8% | 15.1% | 15.6% | 14.7% | **14.1%** | 14.4% | 14.1% |
| Translation (BLEU ↑) | Fr→ En | 33.3 | 33.3 | 34.1 | 33.9 | 34.5 | 34.8 | **35.0** | 35.1 |
| | De→ En | 33.2 | 33.4 | 34.2 | 34.7 | 34.7 | 35.1 | **35.4** | 35.2 |
| | Es→ En | 37.3 | 37.4 | 37.9 | 38.2 | 38.6 | 39.0 | **39.1** | 39.0 |
| | Ca→ En | 28.8 | 28.9 | 29.5 | 29.8 | 30.3 | 30.8 | **31.1** | 30.9 |
| | Ave. | 33.1 | 33.3 | 33.9 | 34.0 | 34.5 | 34.9 | **35.2** | 35.1 |

weighting increases memory usage from 12.5 GB to 14.7 GB (∼18%) and extends each epoch by 0.6 hours, from 3.5 h to 4.1 h (∼17%), due to additional gradient computations. In contrast, the efficient MSP variant—which restricts dynamic weight calculations to only the layers exhibiting gradient conflicts—limits overhead to just 0.3 GB and 0.2 h per epoch while preserving performance (see Appendix D). Moreover, our MSP framework scales gracefully with more tasks, reducing deployment resource demands by consolidating all objectives into a single model (see Appendix I).

Based on our experiments, we summarize our observations in the following sections.

## 4.1 Conflicting speech recognition and translation objectives

Presence of multiple conflicting objectives degrades the model's performance. In this section, we investigate the effect of conflicting objectives on model performance using two algorithmic settings—pre-training+fine-tuning and VS-MSP—across different model sizes. In Figure 3, we present the cosine similarities between the supervised gradient vectors for five languages and the self-supervised gradient used in speech recognition and translation. The accompanying heat map visualizes these similarity scores, highlighting which objectives align strongly (high similarity) and which diverge. As shown, objectives with lower similarity measures tend to conflict more intensely, whereas the self-supervised gradients exhibit markedly higher alignment with the other objectives. We further investigate the presence of conflicting objectives in MSP in Appendix E.

Table 4: Speech recognition and translation on CoVoST-2 using a Conformer model with penalty increase rates of 0.002 vs. 0.02 per epoch.

| Task | Lang / Lang→Eng | Two-stage training | VM-MSP UAS (IR=.02) | VM-MSP USA (IR=.02) | VM-MSP UAS (IR=.002) | VM-MSP USA (IR=.002) |
|---|---|---|---|---|---|---|
| Speech recognition (83 M) (WER ↓) | En | 28.6% | 26.9% | 27.2% | 28.9% | **26.3%** |
| | Fr | 26.1% | 24.1% | 24.3% | 26.4% | **23.8%** |
| | De | 22.5% | 21.2% | 21.5% | 22.8% | **20.5%** |
| | Es | 23.9% | 22.1% | 22.3% | 24.1% | **21.7%** |
| | Ca | 25.2% | 23.8% | 24.1% | 25.5% | **23.2%** |
| | Ave. | 25.3% | 23.6% | 23.9% | 25.5% | **23.1%** |
| Translation (83 M) (BLEU ↑) | Fr→En | 24.2 | 26.5 | 26.6 | **26.9** | 24.9 |
| | De→En | 22.4 | 24.5 | 24.7 | **25.3** | 22.8 |
| | Es→En | 24.6 | 26.1 | 26.5 | **26.7** | 25.3 |
| | Ca→En | 19.7 | 21.3 | 21.6 | **22.1** | 20.1 |
| | Ave. | 22.7 | 24.6 | 24.9 | **25.3** | 23.3 |

As shown in Table 2 and Table 3, the VS-MSP method consistently outperforms the two-stage method. The key difference between these two approaches is the use of multi-objective optimization. This result suggests the presence of conflicts among the speech recognition and translation objectives and highlights the effectiveness of multi-objective optimization in addressing these conflicts.

## 4.2 Enhancing performance with multilevel optimization

Multilevel optimization significantly improves MSP performance by effectively balancing learning objectives and narrowing the search for optimal Pareto stationary points. This section examines the impact of multilevel optimization on MSP performance. We tested two optimization sequences: UAS (self-supervised → speech recognition → translation) and USA (self-supervised → translation → recognition). In the UAS sequence, the unsupervised loss is optimized first, followed by the ASR objective, and then the translation objective. Conversely, for the USA sequence, the unsupervised objective is optimized first, followed by the translation objective and then the ASR one.

SPEECH RECOGNITION: Tables 2 and 3 show that VM-MSP consistently achieves the lowest WER across all languages and model sizes. On the 150 M-parameter Conformer, VM-MSP (USA) reduces WER by 10.2% versus two-stage training and by 4.2% versus VC-MSP. Likewise, on Whisper, VM-MSP (UAS) outperforms two-stage training by 11.3% and VC-MSP by 4.1%. These gains confirm the effectiveness of VM-MSP's multilevel optimization.

TRANSLATION: Tables 2 and 3 show VM-MSP has the highest BLEU scores. On the 150 M-parameter Conformer, VM-MSP (USA) boosts BLEU by 9.1% over two-stage training and 2.8% over VC-MSP. On Whisper, it gains 6.4% over two-stage and 2.0% over VC-MSP. These results underscore VM-MSP's robustness in translation tasks. We observe similar performance gains with the smaller Conformer model and other models[3]

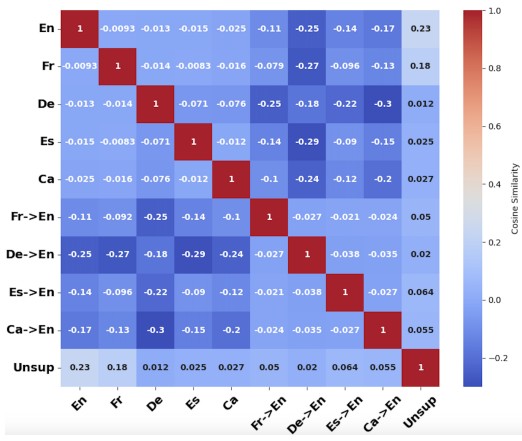

Figure 3: Heat-map of Cosine similarities among ASR and speech translation objectives.

## 4.3 Optimization order in multilevel optimization

The order of optimization impacts MSP accuracy in Multilevel optimization. In this section, we investigate the significance of optimization order in multilevel optimization for MSP. By comparing the performance of different MSP algorithms under varying optimization sequences (UAS and USA), we aim to elucidate how the order of optimization affects MSP performance.

From the results in Table 2 and Table 3, we observe that the UAS optimization sequence consistently yields superior recognition performance compared to USA. This finding indicates the importance of

---

[3]Additional results using BEST-RQ and Wav2Vec2 are presented in Appendix H (Tables 8, 9).

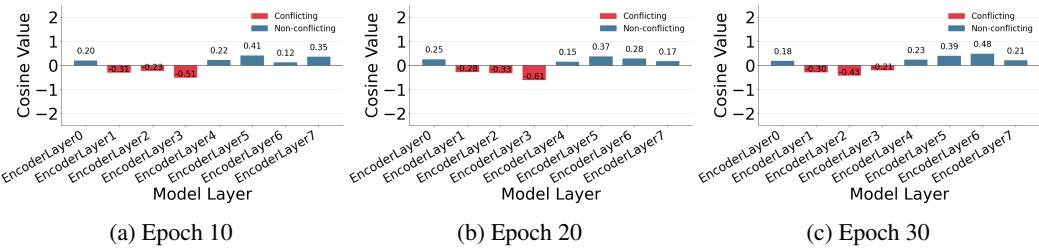

Figure 4: During two-stage training (without multi-objective optimization), the Conformer model exhibits persistent gradient conflicts in the same layers.

prioritizing certain objectives during the training process and highlights the crucial role of optimization order when designing multilevel optimization algorithms for MSP.

## 4.4 Effect of penalty parameter

Penalty parameters play a crucial role in multilevel MSP training. In penalty-based multilevel problems, selecting the appropriate penalty parameter is crucial. These methods prioritize upper-level objectives while controlling lower-level objectives through a penalty term. Using a smaller penalty parameter can weaken constraint enforcement, causing suboptimal

Table 5: Resource comparison per epoch

| Model | GPU Memory (GB) | Time (h/epoch) |
|---|---|---|
| Two-stage | 12.5 | 3.5 |
| MSP (dynamic weighting) | 14.7 | 4.1 |
| MSP (Efficient) | 12.8 | 3.7 |

lower-level performance, slower convergence, and imbalanced optimization [49]. This is evident in our MSP experiments. We further conducted experiments following the same training procedure as other simulations, using a 83M parameter model with two different penalty parameter increase rates. A lower increase rate of 0.002, capped at 1.5, resulted in worse WER for lower-level tasks, as shown in Table 4. Given the equal importance of speech recognition and translation objectives in our study, we applied a larger penalty parameter increase rate of 0.02 for the lower levels, with a final value capped at 1.5. This adjustment improved lower-level performance but slightly degraded upper-level performance. Therefore, selecting the penalty parameter requires careful consideration of the trade-offs between upper- and lower-level priorities. A detailed explanation of this selection process is provided in Appendix H.

## 4.5 Our observations across different model sizes

Our observations are consistent across different model sizes. We assess the consistency of our observations across different model sizes. Results from Tables 2 and 3 confirm the reliability and generalizability of our findings, offering insights for scalable ASR system design.

SPEECH RECOGNITION. From Tables 2 and 3, we observed that the two-stage approach achieved competitive performance across all languages. The VS-MSP method consistently outperformed the two-stage method, and the VC-MSP model demonstrated even better performance. The most notable finding, however, is the performance of the VM-MSP, which exhibited significant improvements. Specifically, the VM-MSP model optimized with the UAS objective sequence achieved the lowest average WER, demonstrating its effectiveness in leveraging unlabeled data. These observations hold true for both the Whisper and Wav2Vec2 models.

TRANSLATION. Tables 2 and 3 illustrate the translation comparison for different models. Similar to speech recognition findings, the two-stage approach demonstrated competitive performance across all language pairs. The VS-MSP model consistently outperformed other algorithms in the translation task. Interestingly, the VM-MSP model optimized with the USA objective sequence achieved the highest average BLEU, outperforming other algorithms.

## 4.6 Consistent observations in language-based multilevel optimization

To verify our findings, we conducted experiments in both task-based and language-based multilevel settings using Conformer models (100 M and 58 M parameters) with comparable hyperparameters

on the LibriSpeech and AISHELL datasets. For the language-based multilevel MSP evaluation, we combined both datasets to focus exclusively on speech recognition. The results (Appendix Table 7) exhibit the same pattern as our task-based optimization experiments, further confirming the effectiveness of the proposed algorithms.

### 4.7 Efficient calculation of the dynamic update direction

Focusing only on the gradients of conflicting layers when computing the update direction improves efficiency without degrading performance. Our layer-wise analysis (Figure 5) reveals a stable pattern: only 20–30% of backbone layers—primarily the early encoder blocks that encode low-level acoustic cues—consistently display large negative cosine similarities, whereas deeper blocks, which model higher-level and more language-agnostic abstractions, exhibit no conflict. Leveraging this observation—and in line with [50]—we apply a lightweight conflict-based pruning strategy that, after detecting the conflicting layers in the first 20 epochs, limits the dynamic update-direction computation to their gradients. This selective scheme shrinks GPU memory from 14.7 GB to 12.8 GB and shortens training time from 4.1 h to 3.7 h per epoch (Table 5) while preserving final performance (see VM-MSP (Efficient) in Tables 2 and 3). Full implementation details are provided in Appendix D.

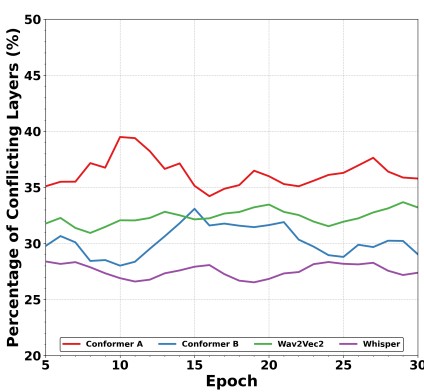

Figure 5: Percentage of conflicting layers.

## 5  Conclusions and Limitations

In conclusion, our study highlights the substantial advantages of integrating self-supervised loss as a constraining objective within a multilevel multi-objective optimization structure for multilingual multi-task speech processing. Our findings indicate that segregating highly conflicting objectives into different optimization levels yields significant benefits for speech recognition and translation tasks. This approach not only enhances the effectiveness of multi-objective optimization but also underscores its potential for optimizing complex tasks across diverse linguistic boundaries.

Our evaluation focused on architectures that exhibit a clear separation between shared and task-specific parameters—for example, a shared encoder followed by distinct classification heads for ASR and translation. This architectural pattern is common in speech processing, but how the method generalizes to more tightly coupled architectures—such as models with a unified decoder shared across tasks—remains an open question. In such cases, task-specific conflicts may propagate more deeply through shared components, potentially requiring alternative optimization strategies or additional regularization. Investigating these scenarios is a promising direction for future research.

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

# Supplementary Materials

## Table of Contents

## A  Detailed Related Work

In this section, we provide a comprehensive review of existing works on multi-objective optimization, multilingual speech recognition, translation, and multi-task learning for speech processing.

**Multi-objective optimization** aims to optimize multiple, often conflicting, objectives simultaneously and has been applied in various domains including engineering [12], finance [18], and decision making [17]. Multi-objective optimization has also been used for solving lexicographic multi-objective problems [38, 51] and speech processing. In previous work, multi-objective optimization has been applied at the system development level, using evolutionary strategies to jointly optimize recognition accuracy and model size by tuning meta-parameters such as model topology and training configuration [40]. In contrast, our work focuses on loss-level conflicts during the training of multilingual and multitask models, where multiple learning signals interact within a shared backbone.

**Multilingual speech recognition and translation.** Early studies in multilingual ASR employed deep neural networks, hidden Markov models, and multilayer perceptrons [27, 54, 56, 22]. The introduction of LSTM models brought significant improvements to multilingual speech recognition [24, 62]. Subsequently, Seq2Seq models with hybrid attention/CTC algorithms and transformer-based architectures emerged as state-of-the-art [58, 55, 61]. For multilingual translation, transformer-based models with self-supervised pre-training have been widely adopted [35, 6, 45]. However, existing approaches often lack the capability to effectively perform multi-tasking across diverse speech-related tasks. These efforts remain orthogonal to our proposed multi-objective optimization algorithms and can potentially benefit from their integration.

**Multi-task Learning for Speech Recognition.** Multi-task learning for joint speech recognition and translation has been explored with limited success. Early attempts introduced algorithms for

| Notation | Description |
|---|---|
| $\Theta \in \mathbb{R}^q$ | Model parameter including backbone and classification head parameter. |
| $\theta \in \mathbb{R}^s$ | Backbone parameter. |
| $\theta^k \in \mathbb{R}^s$ | Backbone parameter at $k$-th iteration. |
| $\theta^* \in \mathbb{R}^s$ | Optimum backbone parameter. |
| $\phi \in \mathbb{R}^r$ | Parameter of the task-specific classification head. |
| $\phi_{t,n} \in \mathbb{R}^r$ | Classification head parameter of $n$-th task and $t$-th language. |
| $\phi_{t,n}^k \in \mathbb{R}^r$ | Classification head parameter of $n$-th task and $t$-th language at $k$-th iteration. |
| $\phi_p \in \mathbb{R}^r$ | A group of all classification head parameters of level $p$. |
| $\phi^* \in \mathbb{R}^r$ | Optimum parameter of the task-specific classification head. |
| $\mathcal{L}$ | Vector of all objectives. |
| $\mathcal{L}_\eta$ | Vector of all objectives with penalized lower-level constrained objective used for VC-MSP method. |
| $\mathcal{L}_p$ | Vector of all objectives in level $p$ used for VM-MSP method. |
| $\ell_m, m \in [M]$ | $m$-th objective. |
| $\ell_s$ | supervised loss with supervised data. |
| $\ell_u$ | self-supervised loss. |
| $t \in [T]$ | Represents a specific language (For example: English, German, etc.). |
| $n \in [N]$ | Represents a specific task (For example: speech recognition or translation.). |
| $k \in [K]$ | Current iteration number. |
| $p \in [P]$ | Optimization level. |
| $\varepsilon$ | Constraint defines the feasible region for the upper-level objectives |
| $d$ | Conflict-avoiding update direction. |
| $\gamma$ | Learning rate of $\lambda$ update. |
| $\alpha$ | Learning rate of backbone parameter. |
| $\beta$ | Learning rate of task-specific classification parameter. |
| $\lambda$ | Dynamic weight to combine the gradient. |
| $\lambda^k$ | Dynamic weight at $k$-th iteration. |
| $\lambda_u^k$ | Dynamic weight of self-supervised objective at $k$-th iteration. |
| $\lambda_{t,n}^k$ | Dynamic weight of $n$-th task and $t$-th language at $k$-th iteration. |
| $\lambda^*$ | Optimum dynamic weight to combine the gradient. |
| $\eta_{p-1}, p \geq 2$ | Penalty parameter of $p$-th level of multilevel optimization (VM-MSP). |
| $\eta = \eta_p \times \eta_{p-1}$ | Combined penalty constant for the lowest level (VM-MSP). |
| $\zeta^k$ | Stochastic unlabeled sample during training at iteration $k$. |
| $\xi^k$ | Stochastic labeled sample during training at iteration $k$. |
| $D$ | Labeled dataset. |

Table 6: List of notations used in this paper

joint speech recognition and translation decoding [1] and intermediate word embeddings with two-stage models [15, 52]. Transformer-based dual encoder-decoder architectures were also developed, featuring separate decoders for speech recognition and translation tasks [33]. Whisper [43] demonstrated the potential of large-scale multitask models, while Mu$^2$SLAM [13] leveraged cross-modality learning across multilingual and supervised subtasks. Joint pre-training and fine-tuning approaches have also been proposed to simplify training [5, 47, 53]. Despite these advancements, most methods rely on static weighting strategies or constrained optimization, which do not explicitly address the challenges posed by conflicting objectives. This can lead to suboptimal performance when tasks inherently conflict.

# B Algorithm Development

After formalizing three training algorithms in Section 3, our subsequent objective is to devise a gradient-based algorithm capable of addressing large-scale, high-dimensional multilingual multi-task

challenges by providing an update rule that converges to Pareto-stationary solutions. We will focus on the algorithm development of VC-MSP, as this can be easily extended to the other two methods (VS-MSP, VM-MSP). To achieve a gradient-based algorithm for VC-MSP that can avoid conflicting update directions, we leverage recent advances in unconstrained multi-objective optimization [9] and employ a penalty-based approach to convert the constrained multi-objective optimization problem in 8 into an unconstrained multi-objective optimization problem. This approach simultaneously conducts self-supervised pre-training and supervised multi-objective learning, as in (8); that is,

$$\min_{\theta \in \mathbb{R}^s, \phi \in \mathbb{R}^r} \mathcal{L}_\eta(\Theta) := [\ell_s(\theta, \phi_{1,1}) + \eta\ell_u(\theta), \cdots, \ell_s(\theta, \phi_{1,N}) + \eta\ell_u(\theta), \ldots, \qquad (12)$$

$$\ell_s(\theta, \phi_{T,1}) + \eta\ell_u(\theta), \cdots, \ell_s(\theta, \phi_{T,N}) + \eta\ell_u(\theta)]$$

where $\eta$ is a penalty parameter. This penalty parameter integrates the self-supervised constrained objective with the supervised objectives and ensures that the feasible region of the supervised objective remains within certain bounds.

**Parameters update.** Using the dynamic weighting and penalization method, we update the backbone parameters, $\theta$, of the MSP model. Next, we describe the backbone parameters and task-specific classification parameters update rules for VS-MSP, VC-MSP, and VM-MSP.

**VS-MSP.** For single vectorized objective training, we only need to consider if the objectives have conflicting update directions. As in multilingual multi-task training, we are using separate language datasets; we can assume that the objectives have conflicting update directions. We can also prove this assumption by calculating $\langle \nabla_\Theta \ell_{t,n}(\Theta), \nabla_\Theta \ell_{t',n'}(\Theta) \rangle < 0$. We optimize these vectorized objectives using the algorithm: 1 where the shared backbone parameters are updated via

$$\theta^{k+1} = \theta^k - \alpha \sum_{t=1}^{T} \sum_{n=1}^{N} \lambda_{t,n}^k \nabla_\theta \ell_s(\theta^k, \phi_{t,n}^k) - \alpha\lambda_u \nabla_\theta \ell_u(\theta^k). \qquad (13)$$

In this context, $\alpha > 0$ denotes the learning rate specifically assigned to the backbone parameters. Moreover, $\lambda_{t,n}^k$ and $\lambda_u$ represent the dynamic update directions for supervised and self-supervised objectives, respectively, which are computed using the MoDo algorithm. Similarly, taking the gradients of each of the supervised objective functions with respect to the parameters of task-specific output heads, task-specific output layers are updated via,

$$\phi_{t,n}^{k+1} = \phi_{t,n}^k - \beta \nabla_\phi \ell_s(\phi_{t,n}^k, \theta^k) \qquad (14)$$

where $\beta > 0$ is the learning rate for the task-specific parameter.

**VC-MSP.** To train a model using the VC-MSP algorithm, the backbone parameters $\theta$ are updated via

$$\theta^{k+1} = \theta^k - \alpha \sum_{t=1}^{T} \sum_{n=1}^{N} \lambda_{t,n}^k \nabla_\theta \ell_s(\theta^k, \phi_{t,n}^k) - \alpha\eta \nabla_\theta \ell_u(\theta^k). \qquad (15)$$

To update task-specific classification heads, we employ (14); see a summary in Algorithm 2.

**VM-MSP.** In VM-MSP, we separate highly conflicting objectives into distinct optimization levels. Here, we assume that all objectives at level $p$ function as lower-level objectives for those at level $p - 1$. Consequently, we can update the backbone parameters using the penalize method, that is

$$\theta^{k+1} = \theta^k - \alpha \sum_{t_1=1}^{T_1} \sum_{n_1=1}^{N_1} \lambda_{t_1,n_1}^k \nabla_\theta \ell_s(\theta^k, \phi_{t_1,n_1}^k) - \alpha\eta_2 \left( \sum_{t_2=1}^{T_2} \sum_{n_2=1}^{N_2} \lambda_{t_2,n_2}^k \nabla_\theta \ell_s(\theta^k, \phi_{t_2,n_2}^k) + \cdots \qquad (16) \right.$$

$$\left. \alpha\eta_{p-1} \left( \sum_{t_p=1}^{T_p} \sum_{n_p=1}^{N_p} \lambda_{t_p,n_p}^k \nabla_\theta \ell_s(\theta^k, \phi_{t_p,n_p}^k) + \cdots \alpha\eta_{P-1} \left( \sum_{t_P=1}^{T_P} \sum_{n_P=1}^{N_P} \lambda_{t_P,n_P}^k \nabla_\theta \ell_s(\theta^k, \phi_{t_P,n_P}^k) + \alpha\eta \nabla_\theta l_u(\theta^k) \right) \right) \right).$$

Update task-specific classification parameters using

$$\phi_{t_p,n_p}^{k+1} = \phi_{t_p,n_p}^k - \beta \nabla_\phi \ell_s(\phi_{t_p,n_p}^k, \theta^k) \qquad (17)$$

where $N_p$ and $T_p$ represent the total number of tasks and languages at level $p$, respectively. We represent the penalty parameter at level $p$ as $\eta_p$, and that for the self-supervised objective as $\eta$.

# C   Task Specific Formulation and Update Rule

In this section, we will explore in detail the three multi-objective optimization setups in speech recognition and translation tasks and establish the parameter update rules for each of them.

## C.1   VS-MSP for single vectorized objectives

For single vectorized objective training, we only need to consider if the objectives have conflicting update directions. As in the multilingual multi-task training, we use separate language datasets, so we can assume that the objectives have conflicting update directions. We can also verify this assumption by calculating $\langle \nabla_\Theta \ell_{t,1}(\Theta), \nabla_\Theta \ell_{t',2}(\Theta) \rangle < 0$. We can formulate this single vectorized objective for ASR and translation tasks following (5) as

$$\min_{\Theta \in \mathbb{R}^q} \; [\ell_s(\theta, \phi_{1,1}), \cdots, \ell_s(\theta, \phi_{1,N}), \ldots, \ell_s(\theta, \phi_{T,1}), \cdots, \ell_s(\theta, \phi_{T,N}), \ell_u(\theta)]. \qquad (18)$$

As there is no lower-level constraint, we optimize this vectorized objective using Algorithm 1, where the shared backbone parameters are updated using the following equations

$$\theta^{k+1} = \theta^k - \alpha \sum_{t=1}^T \lambda_{t,1}^k \nabla_\theta \ell_s(\theta^k, \phi_{t,1}^k) - \alpha \sum_{t=1}^T \lambda_{t,2}^k \nabla_\theta \ell_s(\theta^k, \phi_{t,2}^k) - \alpha \lambda_u^k \nabla_\theta \ell_u(\theta^k) \qquad (19)$$

where $\lambda_{t,1}$ and $\lambda_{t,2}$ are dynamic update directions for ASR and translation tasks, respectively, and $\lambda_u$ is the dynamic update direction for the self-supervised objective calculated using the MoDo algorithm. We update the classification heads using

$$\phi_{t,1}^{k+1} = \phi_{t,1}^k - \beta \nabla_\phi \ell_s(\phi_{t,1}^k, \theta^k). \qquad (20a)$$

$$\phi_{t,2}^{k+1} = \phi_{t,2}^k - \beta \nabla_\phi \ell_s(\phi_{t,2}^k, \theta^k). \qquad (20b)$$

## C.2   VC-MSP for vectorized objectives with constraint lower level

In this setup, we use self-supervised CPC loss, $\ell_u(\theta)$, as a lower-level constraint to guide optimization toward a region with good representation and at the same time admits common descent directions for supervised loss, $\ell_s(\theta, \phi)$. The problem formulation for VC-MSP in speech recognition and translation tasks can be written as follows:

$$\min_{\Theta \in \mathbb{R}^q} \; [\ell_s(\theta, \phi_{1,1}), \ell_s(\theta, \phi_{1,2}), \ldots, \ell_s(\theta, \phi_{T,1}), \ell_s(\theta, \phi_{T,2})]$$

$$\text{s.t. } \ell_u(\theta) - \min_\theta \ell_u(\theta) \leq \varepsilon. \qquad (21)$$

The backbone parameters $\theta$ is updated using,

$$\theta^{k+1} = \theta^k - \alpha \sum_{t=1}^T \lambda_{t,1}^k \nabla_\theta \ell_s(\theta^k, \phi_{t,1}^k) - \alpha \sum_{t=1}^T \lambda_{t,2}^k \nabla_\theta \ell_s(\theta^k, \phi_{t,2}^k) - \alpha \eta \nabla_\theta \ell_u(\theta^k). \qquad (22)$$

The task specific classification parameters are updated using (20a) and (20b)

## C.3   VM-MSP for multilevel ASR optimization

In a multilevel optimization problem, there is a hierarchy of objectives. We can reformulate the multilingual multi-task ASR optimization task into different multilevel optimization problems based on the tasks, languages, or language families to which they belong. We study these set-ups and solve these optimization problems using a penalty-based gradient descent method.

**Multilevel optimization based on tasks.** We can extend the MSP optimization problem into three levels based on the tasks: speech recognition, translation, and self-supervised task. We always place the self-supervised objective at the lowest level and optimize it first, as the optimization of all other

objectives directly depends on the optimization of the self-supervised objective.

$$\underset{\phi_{1,1},\cdots,\phi_{T,1}\in\mathbb{R}^r,\phi_{1,2}^*,\ldots,\phi_{T,2}^*,\theta^*}{\operatorname{argmin}} \mathcal{L}_{\mathrm{s}}(\phi_{1,1},\phi_{2,1},\ldots,\phi_{1,2}^*,\phi_{2,2}^*,\cdots,\theta^*)$$

$$\text{s.t. } \phi_{1,2}^*,\cdots,\phi_{T,2}^* = \underset{\phi_{1,2},\cdots,\phi_{T,2}\in\mathbb{R}^r,\theta^*}{\operatorname{argmin}} \mathcal{L}_{\mathrm{s}}(\phi_{1,1},\phi_{2,1},\ldots,\phi_{1,2},\phi_{2,2},\cdots,\theta^*)$$

$$\text{s.t. } \theta^* = \underset{\theta\in\mathbb{R}^s}{\operatorname{argmin}} \; \ell_{\mathrm{u}}(\theta).$$

(23)

We apply a penalty-based method to convert this multilevel multi-objective optimization problem into a single-level optimization problem and apply dynamic multi-objective optimization to update the parameters in a conflict-avoiding direction.

$$\theta^{k+1} = \theta^k - \alpha\sum_{t=1}^{T}\lambda_{t,1}^k\nabla_\theta\ell_{\mathrm{s}}(\theta^k,\phi_{t,1}^k) - \alpha\eta_1\left(\sum_{t=1}^{T}\lambda_{t,2}^k\nabla_\theta\ell_{\mathrm{s}}(\theta^k,\phi_{t,2}^k) + \alpha\eta_2\nabla_\theta\ell_{\mathrm{u}}(\theta^k)\right). \quad (24)$$

Here, $\eta_1$ and $\eta_2$ are penalty parameters. We can combine $\eta_1$ and $\eta_2$ and get $\eta = \eta_1 \times \eta_2$ for self-supervised loss.

$$\theta^{k+1} = \theta^k - \alpha\sum_{t=1}^{T}\lambda_{t,1}^k\nabla_\theta\ell_{\mathrm{s}}(\theta^k,\phi_{t,1}^k) - \alpha\eta_1\sum_{t=1}^{T}\lambda_{t,2}^k\nabla_\theta\ell_{\mathrm{s}}(\theta^k,\phi_{t,2}^k) - \alpha\eta\nabla_\theta\ell_{\mathrm{u}}(\theta^k). \quad (25)$$

Next, we update the classification heads via

$$\phi_{t,1}^{k+1} = \phi_{t,1}^k - \beta\nabla_\phi\ell_{\mathrm{s}}(\phi_{t,1}^k,\theta^k). \quad (26a)$$

$$\phi_{t,2}^{k+1} = \phi_{t,2}^k - \beta\nabla_\phi\ell_{\mathrm{s}}(\phi_{t,2}^k,\theta^k). \quad (26b)$$

We provide a detailed algorithm of multilevel ASR optimization in 3. We also do experiments altering the optimization order of ASR and translation tasks.

**Multilevel optimization based on language.** We can also extend the ASR optimization problem to multiple levels based on languages.

$$\underset{\phi_{1,1},\phi_{1,2}\in\mathbb{R}^r,\phi_{2,1}^*,\phi_{2,2}^*,\ldots,\theta^*}{\operatorname{argmin}} \mathcal{L}_{\mathrm{s}}(\phi_{1,1},\phi_{1,2},\phi_{2,1}^*,\phi_{2,2}^*,\cdots,\theta^*)$$

$$\ddots$$

$$\text{s.t. } \phi_{T,1}^*,\phi_{T,2}^* = \underset{\phi_{T,1},\phi_{T,2}\in\mathbb{R}^r,\theta^*}{\operatorname{argmin}} \mathcal{L}_{\mathrm{s}}(\phi_{1,1},\phi_{1,2},\ldots,\phi_{T,1},\phi_{T,2},\theta^*)$$

$$\text{s.t. } \theta^* = \underset{\theta\in\mathbb{R}^s}{\operatorname{argmin}} \; \ell_{\mathrm{u}}(\theta).$$

(27)

In this setup, we optimize all the objectives of one language in one optimization level and optimize other languages' objectives in other optimization levels. For simplicity of implementation, we will consider two languages. We can update the model parameters using the following penalty-based update rules.

$$\theta^{k+1} = \theta^k - \alpha\sum_{n=1}^{N}\lambda_{1,n}^k\nabla_\theta l_{\mathrm{ctc}}(\theta^k,\phi_{1,n}^k) - \alpha\eta_1\sum_{n=1}^{N}\lambda_{2,n}^k\nabla_\theta l_{\mathrm{ctc}}(\theta^k,\phi_{2,n}^k) - \alpha\eta\nabla_\theta l_{\mathrm{u}}(\theta^k). \quad (28)$$

In this equation, $\eta_1$ and $\eta_2$ are penalty parameters. We can combine $\eta_1$ and $\eta_2$ to obtain $\eta = \eta_1 \times \eta_2$, which is used for the self-supervised loss. The parameter $N = 2$ represents the total number of tasks (in this experiment, ASR and translation). The terms $\lambda_{1,n}^k$ and $\lambda_{2,n}^k$ represent the dynamic update directions for languages 1 and 2, respectively, during the $k$-th iteration for task $n$.

Next, we update the classification heads via

$$\phi_{t,1}^{k+1} = \phi_{t,1}^k - \beta\nabla_\phi\ell_{\mathrm{s}}(\phi_{t,1}^k,\theta^k). \quad (29a)$$

$$\phi_{t,2}^{k+1} = \phi_{t,2}^k - \beta\nabla_\phi\ell_{\mathrm{s}}(\phi_{t,2}^k,\theta^k). \quad (29b)$$

In both task-based and language-based MLO, we alter the order of objectives at the optimization level to examine the effects of their arrangement. By doing so, we can better understand how the sequence of objectives influences the optimization process and outcomes.

## D    Efficient Training

To reduce computational overhead during dynamic update-direction computation, we introduce a lightweight method for identifying conflicting layers. For each shared encoder layer $l$, we compute task-specific gradients $\mathcal{G}_i^{(l)}$ and evaluate their pairwise cosine similarities. The cosine similarity between gradients of tasks $i$ and $j$ at layer $l$ is defined as:

$$\cos \theta_{ij}^{(l)} = \frac{\langle \mathcal{G}_i^{(l)}, \mathcal{G}_j^{(l)} \rangle}{\|\mathcal{G}_i^{(l)}\| \cdot \|\mathcal{G}_j^{(l)}\|}.$$

We denote the set of conflicting task pairs at layer $l$ as $\mathcal{P}^{(l)} := \{(i,j) \mid \cos \theta_{ij}^{(l)} < 0\}$. A layer is considered to be conflicting if the average cosine similarity across all such pairs is negative:

$$\frac{1}{|\mathcal{P}^{(l)}|} \sum_{(i,j) \in \mathcal{P}^{(l)}} \cos \theta_{ij}^{(l)} < 0.$$

Only layers meeting this criterion are selected for computing the conflict-avoidance update direction $d$. This targeted layer selection substantially reduces training time and memory usage (see Section 4.7 and Table 5), while maintaining the performance gains of multilevel optimization.

## E    Gradient Conflict

In this setup, we aim to separate highly conflicting objectives into upper and lower optimization levels. However, a sub-question arises within this setup: which objectives are highly conflicting? To address this question, we need to establish a boundary or threshold that distinguishes objectives with significant conflicts. We can create such a threshold by calculating the degree of conflict using the cosine similarity of the gradients of the objectives. If the cosine similarity of two objectives is smaller than a certain threshold, they are optimized at different levels. If $\nabla_\Theta \ell_{t,n}(\Theta)$ and $\nabla_\Theta \ell_{t',n'}(\Theta)$ are gradients of two objectives, then we can calculate the cosine similarity using

$$\cos \omega = \frac{\langle \nabla_\Theta \ell_{t,n}(\Theta), \nabla_\Theta \ell_{t',n'}(\Theta) \rangle}{\|\nabla_\Theta \ell_{t,n}(\Theta)\| \|\nabla_\Theta \ell_{t',n'}(\Theta)\|} \tag{30}$$

where $\omega$ is the angle between the gradients of two different objectives. To calculate the similarity between update directions, we use the same conformer model and train it using two different languages and objectives simultaneously. We train the model for 20 epochs using both objectives and then average the gradients of their updates separately. We follow the same process for all languages and record their average gradients for 20 epochs. We can now calculate the cosine similarity between the gradient update direction of two objectives from these recorded gradients. We also compare the cosine similarity between self-supervised and supervised losses.

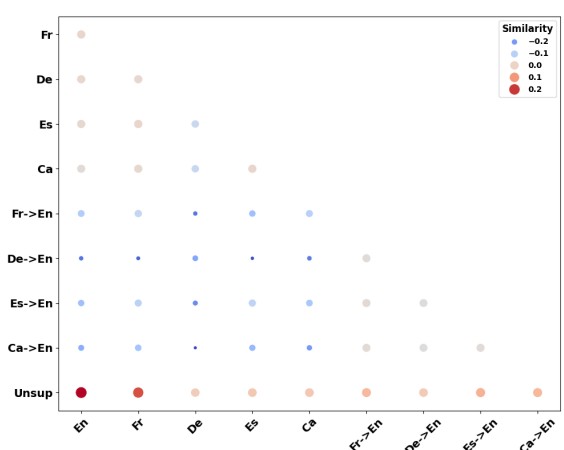

Figure 6: Scatter plot of cosine similarities between MSP objectives.

In Figure 3 and 6, we depict the cosine similarity of supervised objective gradients across five languages, along with the self-supervised objective gradient for speech recognition and translation. The heat map displays the similarity values, while the scatter plot, with points colored by their cluster assignments, helps visualize which objectives are closely related (high similarity) and which are not. The size and color of the points represent the similarity values and cluster assignments, respectively.

---

**Algorithm 1** VS-MSP for multilingual multi-task MSP.

---

**Input:** Labeled data $(x, y)$, unlabeled data $X_{\mathrm{u}} \coloneqq \{x_{\mathrm{u}}^1, x_{\mathrm{u}}^2, \cdots, x_{\mathrm{u}}^E\}$, learning rates $\alpha$ and $\beta$;
**for** $k = 1$ **to** $K$ **do**
    sample $\zeta_1^k = x_{1,\mathrm{u}}^k, \zeta_2^k = x_{2,\mathrm{u}}^k, \xi_1^k = (x_1^k, y_1^k)$ and $\xi_2^k = (x_2^k, y_2^k)$
    compute $\nabla \ell_{\mathrm{u}}(\zeta_1^k; \theta^k), \nabla \ell_{\mathrm{u}}(\zeta_2^k; \theta^k), \nabla \ell_{\mathrm{s}}(\xi_1^k; \theta^k, \phi^k), \nabla \ell_{\mathrm{s}}(\xi_2^k; \theta^k, \phi^k)$
    update $\lambda^{k+1}$ by (4)
    update $\theta^{k+1}$ by (13)
    update $\phi_{t,n}^{k+1}$ by (14) $\forall t \in [T], \forall n \in [N]$
**end for**
**Output:** $\theta^K, \{\phi_{t,n}^K\}$

---

---

**Algorithm 2** VC-MSP for multilingual multi-task MSP

---

**Input:** Labeled data $(x, y)$, unlabeled data $X_{\mathrm{u}} \coloneqq \{x_{\mathrm{u}}^1, x_{\mathrm{u}}^2, \cdots, x_{\mathrm{u}}^E\}$, learning rates $\alpha, \beta$, and penalty parameter $\eta$;
**for** $k = 1$ **to** $K$ **do**
    sample $\zeta^k = x_{\mathrm{u}}^k, \xi_1^k = (x_1^k, y_1^k)$ and $\xi_2^k = (x_2^k, y_2^k)$
    compute $\nabla \ell_{\mathrm{u}}(\zeta^k; \theta^k), \nabla \ell_{\mathrm{s}}(\xi_1^k; \theta^k, \phi^k), \nabla \ell_{\mathrm{s}}(\xi_2^k; \theta^k, \phi^k)$
    update $\lambda^{k+1}$ by (4)
    update $\theta^{k+1}$ by (15)
    update $\phi_{t,n}^{k+1}$ by (14) $\forall t \in [T], \forall n \in [N]$
**end for**
**Output:** $\theta^K, \{\phi_{t,n}^K\}$

---

Our analysis of these figures reveals that tasks with lower cosine similarity exhibit higher conflicts. Notably, the self-supervised gradients show significantly higher alignment with the other objectives. We found that segregating the highly conflicting speech recognition and translation tasks into different optimization levels reduced the overall conflict among the objective gradients, which consequently improved the overall performance.

# F    Baseline Training Methods

In this section, we outline the baseline methods used to compare against our multi-objective optimization algorithms.

### F.1    Two-stage training (Pre-training + Fine-tuning)

This method involves two sequential steps:

**1. Pre-training:** The model is first pre-trained on a self-supervised learning objective, such as CPC, Whisper or Wav2Vec2, to learn general-purpose representations from unlabeled speech data. During this stage, the backbone parameters are updated using:

$$\theta^{k+1} = \theta^k - \alpha \nabla_\theta \ell_{\mathrm{u}}(\theta^k), \tag{31}$$

where $\ell_{\mathrm{u}}$ represents the self-supervised loss, and $\alpha$ is the learning rate.

**2. Fine-tuning:** After pre-training, the model is fine-tuned on a supervised task (e.g., speech recognition and translation) using the CTC or cross-entropy loss to adapt the learned representations to task-specific objectives. During fine-tuning:

- The backbone parameters are updated using:

$$\theta^{k+1} = \theta^k - \frac{\beta}{NT} \sum_{t=1}^{T} \sum_{n=1}^{N} \nabla_\theta \ell_{\mathrm{s}}(\theta^k, \phi_{t,n}^k), \tag{32}$$

where $\beta$ is the learning rate, $N$ and $T$ denote the number of tasks and languages, respectively.

**Algorithm 3** VM-MSP for multilingual multi-task MSP.

---

**Input:** Labeled data $(x, y)$, unlabeled data $X_\mathrm{u} := \{x_\mathrm{u}^1, x_\mathrm{u}^2, \cdots, x_\mathrm{u}^E\}$, learning rates $\alpha, \beta$, and penalty $\eta_1, \cdots, \eta_P$;
**for** $k = 1$ **to** $K$ **do**
    sample $\zeta^k = x_\mathrm{u}^k$, $\xi_1^k = (x_1^k, y_1^k)$ and $\xi_2^k = (x_2^k, y_2^k)$
    compute $\nabla \ell_\mathrm{u}(\zeta^k; \theta^k)$, $\nabla \ell_\mathrm{s}(\xi_1^k; \theta^k, \phi^k)$, $\nabla \ell_\mathrm{s}(\xi_2^k; \theta^k, \phi^k)$
    update $\lambda^{k+1}$ by (4)
    update $\theta^{k+1}$ by (16)
    update $\phi_{t_p, n_p}^{k+1}$ by (17)$\forall t_p \in [T_p], \forall n_p \in [N_p]$
**end for**
**Output:** $\theta^K, \{\phi_{t,n}^K\}$

---

- The parameters of the individual classification heads are updated using:

$$\phi_{t,n}^{k+1} = \phi_{t,n}^k - \beta \nabla_\phi \ell_\mathrm{s}(\phi_{t,n}^k, \theta^k), \tag{33}$$

where $\phi_{t,n}$ denotes the parameters for task $n$ and language $t$.

## F.2 Static Weighting

This method follows the same process as pre-training + fine-tuning but introduces static weighting during fine-tuning. Instead of using equal weights for all supervised objectives, a grid search is performed to assign suitable weights to each objective. The backbone parameters are updated using:

$$\theta^{k+1} = \theta^k - \beta \sum_{t=1}^T \sum_{n=1}^N \mu_{t,n} \nabla_\theta \ell_\mathrm{s}(\theta^k, \phi_{t,n}^k), \tag{34}$$

where $\mu_{t,n}$ represents the static weight assigned to the supervised objective for task $n$ and language $t$. For our experiments, the following language-specific weights were used:

$$[\text{En, Fr, De, Es, Ca}] = [0.18, 0.19, 0.27, 0.16, 0.20].$$

## F.3 Joint two-stage training without multi-objective optimization

This method follows the same process as VC-MSP but does not incorporate multi-objective optimization [47]. Instead, all supervised objectives are optimized jointly without dynamic weighting or conflict-aware gradient alignment, resulting in a simpler optimization process.

# G Experimental setup

In this section, we outline the dataset, models, hyperparameters, and data pre-processing techniques employed in evaluating our VS-MSP, VC-MSP, and VM-MSP algorithms.

**Dataset.** We evaluate our training algorithms on a combined dataset of LibriSpeech [42], AISHELL v1 [7], and CoVoST v2 [57]. LibriSpeech is an English speech dataset consisting of 960 hours of data along with transcripts. AISHELL v1 is a 178-hour multi-channel Mandarin speech corpus designed for various speech/speaker processing tasks. We have combined these two datasets to create a single multilingual dataset. Our approach involved splitting the LibriSpeech dataset, allocating 860 hours for self-supervised pre-training and using the 100-hour train-clean-100 subset for supervised training. The trained models are tested on the AISHELL test dataset and the LibriSpeech test-clean dataset. During training using CoVoST dataset, we use equal batch sizes across all languages and tasks to ensure balanced training. For high-resource En, we use the ful data without up-sampling, while applying up-sampling for low-resource languages—4x for Ca and Es and 2x for Fr and De.

In the first experiment, we use a combined LibriSpeech and AISHELL multilingual dataset and train a multi-head conformer for multilingual MSP tasks. In the second experiment, we use the CoVoST v2 training dataset for multilingual speech recognition and translation training. The CoVoST v2 test set is used to evaluate the trained models. CoVoST v2 is a widely used benchmark multilingual

Table 7: ASR WERs (LibriSpeech) and CERs (AISHELL) for two-stage training, Joint training, VS-MSP, VC-MSP, and VM-MSP, with VM-MSP using UEC (self-supervised → English → Chinese) and UCE (self-supervised → Chinese → English) optimization sequences.

| Param | Lang | Two-stage training | Joint training | VS-MSP | VC-MSP | VM-MSP UEC | VM-MSP UCE |
|---|---|---|---|---|---|---|---|
| 100M | En (test-clean) | 6.2% | 5.9% | 6.1% | 5.7% | **5.2%** | 5.4% |
| | En (test-other) | 17.0% | 16.8% | 17.1% | 16.7% | **16.3%** | 16.5% |
| | Zh | 6.0% | 5.6% | 5.8% | 5.5% | 5.3% | **5.0%** |
| | Ave | 9.7% | 9.4% | 9.6% | 9.3% | **8.9%** | **8.9%** |
| 58M | En (test-clean) | 7.8% | 7.1% | 7.3% | 6.8% | **6.5%** | 6.6% |
| | En (test-other) | 17.8% | 17.5% | 17.7% | 17.3% | **17.0%** | 17.1% |
| | Zh | 7.4% | 6.8% | 7.0% | 6.5% | 6.1% | **5.8%** |
| | Ave | 11.0% | 10.4% | 10.6% | 10.2% | 9.9% | **9.8%** |

translation corpus covering translations from 21 languages into English and from English into 15 languages.

**Models and hyper-parameters.** For additional simulations, we employ the wav2vec2 model.

wav2vec2-large + Transformer decoder. We initialize from XLSR-53 (24 layers, $d = 1024$, $H = 16$, FF dim 4096). ASR is handled by a linear CTC head over the transcription vocabulary. For translation, we append a Transformer decoder with three encoder layers and three decoder layers (dim 1024, $H = 16$, FF dim 4096), outputting via a linear projection to the translation vocabulary.

Hyper-parameters. We use grid search to optimize hyperparameters, including learning rate, batch size, step size of MoDo, and penalty parameter increasing rate. For both SSL pre-training and supervised fine-tuning, the backbone learning rate is consistently set higher than the classification parameter learning rate. The SSL pre-training phase starts with a learning rate of $\alpha = 5 \times 10^{-4}$ for 100 epochs, annealed by a factor of 0.1 every 20 epochs. Fine-tuning uses a maximum learning rate of $\beta = 5 \times 10^{-5}$, with a scheduler reducing the learning rate by a factor of 0.1 if the test loss does not improve within 10 epochs. All multi-objective models (VS-MSP, VC-MSP, and VM-MSP) and joint PT+FT models are trained for 200 epochs. For PT+FT, we pre-train the model for 200 epochs and fine-tune it for an additional 100 epochs. A batch size of 256 and AdamW optimizer are used for both self-supervised and supervised training. The same hyperparameter settings are applied across all training methods to ensure consistency and comparability.

Penalty parameter for ASR and translation. For VC-MSP, the initial penalty parameter $\eta$ is set to 0 and increases by 0.02 per epoch. The increase stops once the penalty reaches a maximum value of 1.5. For VM-MSP, the second-level penalty parameter $\eta_1$ is initially set to 0.1 and increases by 0.02 per epoch, while the lower-level penalty constant $\eta_2$ starts at 0 and also increases by 0.02 per epoch. The increase for both penalty constants stops once they reach 1.5. A higher increase rate for the lower level ensures equal importance of both upper-level and lower-level objectives.

Data pre-processing. Our experiments cover both supervised and self-supervised regimes and share a common log-Mel preprocessing pipeline. Raw audio is converted to 80-dimensional log-Mel spectrograms and normalized to zero mean and unit variance. In the SELF-SUPERVISED setup, the model receives a 2 s context window and is trained to predict the following 1 s segment without any data augmentation. In the SUPERVISED setup, we apply SpecAugment to the normalized features to improve robustness. For CONFORMER-BASED MODELS, transcripts are tokenized with SentencePiece [31] using a 1,000-token word-level vocabulary for every language except Chinese, where we employ a 4,930-token character-level model. For WAV2VEC 2.0, we use the character-level CTC vocabulary bundled with the official checkpoints, and for WHISPER we adopt the byte-pair-encoding tokenizer released with its original implementation.

Computational Resources. All simulations were run on two NVIDIA A5000 GPUs and two NVIDIA A4500 GPUs, with an Intel i9-7920X CPU and 128 GB of DDR4 memory.

# H  Ablation Study

In this section, we study the impact of different pre-training methods and provide a detailed explanation of the effect of the penalty parameter on the overall training process.

Table 8: Speech recognition WERs and translation BLEU score comparison between CPC and BEST-RQ pre-training methods. For translation we do Lang → En translation.

| Param | Lang | VM-MSP-UAS (CPC-WER) | VM-MSP-UAS (CPC-BLEU) | VM-MSP-UAS (BEST-RQ-WER) | VM-MSP-UAS (BEST-RQ-BLEU) |
|---|---|---|---|---|---|
| 150M | En | 19.7% | – | **19.1%** | – |
| | Fr | 21.8% | 29.5 | **20.9%** | **30.1** |
| | De | 14.6% | 28.5 | **14.1%** | **29.2** |
| | Es | 17.0% | 31.3 | **16.5%** | **31.9** |
| | Ca | 19.1% | 25.4 | **18.6%** | **26.2** |
| | Ave. | 18.4% | 28.7 | **17.8%** | **29.4** |

Table 9: Speech recognition WERs and translation BLEU scores between Wav2Vec2 with and without VM-MSP methods. For translation, we perform translation from Lang → En.

| Param | Lang | Joint training | | Joint training | |
| | | Wav2Vec2 (WER) Without VM-MSP | Wav2Vec2 (BLEU) Without VM-MSP | Wav2Vec2 (WER) With VM-MSP | Wav2Vec2 (BLEU) With VM-MSP |
|---|---|---|---|---|---|
| 300M | En | 18.1% | – | **16.3%** | – |
| | Fr | 19.3% | 31.2 | **18.2%** | **32.4** |
| | De | 14.0% | 29.9 | **13.1%** | **30.8** |
| | Es | 16.1% | 34.3 | **15.2%** | **35.1** |
| | Ca | 18.9% | 26.2 | **18.0%** | **27.2** |
| | Ave. | 17.2% | 30.4 | **16.1%** | **31.4** |

## H.1 Impact of Pre-training Method

In this ablation study, we assess the impact of two different pre-training techniques—CPC and BEST-RQ [14]—on the performance of our VM-MSP method. The purpose of this ablation is to isolate the contribution of the pre-training method to the overall performance of the MSP tasks. We keep the settings consistent across both methods, with the model containing 150 million parameters in all cases. The tasks evaluated include speech recognition in various languages and translation for translating from different source languages into English.

The results in Table 8 compare CPC and BEST-RQ across five languages. The results indicate a consistent improvement when using the BEST-RQ pre-training method. Specifically, BEST-RQ leads to a 3.3% absolute improvement in the average WER compared to CPC across all languages.

On the translation task, BEST-RQ also outperforms CPC, resulting in a 2.4% absolute increase in the average BLEU score across the evaluated languages. This indicates that BEST-RQ not only improves the speech recognition task but also enhances the downstream translation quality, likely due to the richer representations learned during pre-training.

Overall, these results suggest that the pre-training method plays a crucial role in enhancing both speech recognition and translation performance. The BEST-RQ approach, with its enhanced capability to model complex speech patterns, proves to be more effective than CPC, thus making it the more suitable choice for the VM-MSP algorithm.

## H.2 Impact of VM-MSP on fine-tuning speech foundation model

We evaluate our VM-MSP (UAS) method using the pre-trained Wav2Vec2-XLS-R[4] model [2]. In this approach, we use the pre-trained model as the backbone and add task- and language-specific linear layers and decoder to predict the output vocabulary, training with the CTC and cross-entropy loss. All hyperparameters remain consistent with our previous training protocols, except for the tokenizer. Specifically, we use the same tokenizer as the pre-trained Wav2Vec2 model to maintain consistency in subword segmentation and ensure compatibility with the model's learned representations.

In the first experiment, we adopt joint pre-training+fine-tuning approach, training the Wav2Vec2 model across all languages for speech recognition and translation tasks while following the same procedure as our earlier joint pre-training+fine-tuning training. In the second experiment, we extend joint two-stage with multi-objective optimization, similar to the VM-MSP (USA) training approach. For both experiments, the model is trained for 50 epochs. The results, summarized in Table 9, show

---

[4]https://huggingface.co/facebook/wav2vec2-xls-r-300m

Table 10: Comparison of resource requirements between a single multi-objective model and multiple single-objective models during deployment.

| Model | Encoder Param | Classification Heads | Total Param | Storage Size | Loading Time (s) | Inference time (ms) |
|---|---|---|---|---|---|---|
| MSP Model | ~58M | ~25M | ~83M | ~158.4 MB | ~0.10 | ~11.4 |
| Five Single-Objective Models | ~290M | ~125M | ~415.0M | ~792 MB | ~0.5 | ~55.9 |

that, on average, the Wav2Vec2 model trained with VM-MSP outperforms the standard Wav2Vec2 model by 6.4% in the speech recognition task and by 3.3% in the translation task.

### H.3 Impact of penalty parameter

In our multilingual multi-task ASR experiments, we investigated the effects of different penalty parameter increase rates to balance the ASR and translation tasks. We tested two configurations:

- A **lower increase rate of 0.002**, which led to worse WER/BLEU score for lower-level tasks, as shown in Tables 4.
- A **higher increase rate of 0.02**, which improved lower-level performance but slightly degraded upper-level performance.

Choice of capped value for the penalty parameter: We capped the penalty parameter at 1.5 based on our observed trade-off between upper- and lower-level tasks. A penalty higher than 1.5 could have improved lower-level performance further, but it would have significantly degraded upper-level metrics. Thus, 1.5 was chosen as an optimal balance point.

Post-Maximum Penalty Effects: The penalty parameter reached its maximum value of 1.5 after 75 epochs, but training continued for another 25 epochs. During this time, we observed further improvements in lower-level WER/BLEU scores, while upper-level performance deteriorated. This reinforces the critical role that penalty parameter selection plays in balancing competing objectives.

## I Resource Efficiency of the multi-objective optimization Model

This section addresses the question: **How does a single multi-objective optimization model reduce resource demands during deployment, making it a more efficient solution overall?**

- Reduced Storage Requirements: A single multi-objective model is highly memory-efficient due to parameter sharing across tasks, see Table: 10. For example, the 83 M multi-objective model used in our experiments has a size of 158.4 MB, comprising an encoder (~58M parameters) shared across all objectives and classification heads (~25M parameters). In contrast, deploying five separate models for these tasks would require $5\times$ MORE BACKBONE PARAMETERS, resulting in significantly higher storage demands. Assuming each single-objective model uses an encoder of similar size, the total storage requirement for separate models would reach approximately 792 MB.
- Efficient Inference: The multi-objective model also minimizes latency and computational overhead during inference. In our system, it takes only 0.10s to load the single multi-objective model, whereas loading five separate models takes 0.5s. This reduction in loading time directly translates to faster response times and improved computational efficiency.

By consolidating multiple objectives into a single model, the multi-objective optimization approach not only achieves significant memory savings but also ensures faster deployment and reduced computational demands, making it a scalable and efficient solution for real-world applications.

