# OpenReview forum: "Objective Soups: Multilingual Multi-Task Modeling for Speech Processing"
_NeurIPS.cc/2025/Conference — NeurIPS 2025 poster_

### Official Review · Reviewer_UXqi · 2025-06-04

**Clarity:** 2
**Significance:** 3
**Originality:** 3
**Rating:** 5
**Confidence:** 4

**Summary:**

The paper proposes a method that addresses the multiple objectives when training a speech translation and speech recognition model for a handful of languages. These training objectives are in conflict with each other, the authors propose to remedy this by task/language specific augmentation parameters and alternating between pretraining (language model objective) and task-specific training (ST and ASR).

**Questions:**

N/A

**Ethical Concerns:**

["NO or VERY MINOR ethics concerns only"]

**Final Justification:**

The feedback by the authors cleared up some of my confusion, and I would be happy to see the paper accepted.

**Limitations:**

Yes.

**Paper Formatting Concerns:**

None.

**Quality:**

3

**Strengths And Weaknesses:**

There are several points where the paper is hard to understand and first (to me) suggested to go into different directions than it ultimately did. The paper has a long discussion of conflicting gradients, a definition of "Pareto stationary" solutions. The method is first described by nested minimization (which is hard to understand since after minimization to obtain parameter values, you cannot change them again). But the solution is the introduction of task-specific parameters but otherwise relies on task/language specific weights. The nested minimization in the Figure 2 leads ultimately to the introduction of additional weight ("penalty") parameters. Their role is not very well described, although a key feature is that they increasingly downweight the secondary/tertiary objectives.

There is a very long appendix (which I did not read in detail) that may clear up this confusion - but the main body of the paper should be more clear about these points.

The method has a lot of hyper parameters, such as task/language specific weights, the penalty parameters. However, the selection of these hyper parameters is not described in the main body of the paper.

Based on what the paper ultimately does, it shows clear empirical evidence that the proposed added parameters and training scheme works well. The novelty of the approach, as I understand it, is limited (task-specific augmentation parameters and curriculum training are common techniques), but there is still sufficient substance here for acceptance of the paper.

There is some nice analysis how gradient conflicts evolve over time (Figure 3), although they do not seem to change much during training.

---

> ### Author Rebuttal · Authors · 2025-07-31
>
> We thank the reviewer for appreciating our work. We hope our responses to your comments below address your concerns.
>
> **Q1: Nested minimization is hard to understand; after minimization you cannot change parameters again.**
>
> In the paper, the nested formulation is a conceptual statement of priorities. Operationally, we implement it via a standard penalty‑based conversion to an unconstrained vector optimization (Eq. 9) and then use dynamic multi‑objective updates (MoDo) to obtain a conflict‑avoiding common‑descent direction (Eqs. (10)–(12)), with explicit first‑order update rules summarized in Table 1 and Algorithms 2–3. Thus, no literal “re‑minimization after minimization” occurs during training—the solver performs single‑loop gradient updates with penalties that encode the nesting.
>
>
> **Q2: Role of the penalty (‘additional weight’) parameters is not well described; they increasingly down‑weight lower levels.**
>
> In the updated version of the paper, we clarify the role of the penalties in two places in the main text. First, we explain the $\varepsilon$-constraint perspective: increasing $\eta$ restricts iterates to a representation‑preserving neighborhood $\mathcal{R}_{\delta}$ around the self‑supervised minimizer, which encodes the lower‑level constraint during upper‑level optimization. Second, we give the concrete backbone update rules showing exactly where $\eta$ (and $\eta_1$ at the intermediate level) enters.
>
> **More explanation is provided in reviewer Ljor Q:3 (question regarding Line 120) section**
>
> **Q3: Many hyperparameters; selection not described in the main body.**
>
> We report the core settings—optimizer, learning rates, and (crucially) the penalty schedule $\eta$—directly in Section 4. The detailed hyperparameter‑selection process has been added to Appendix G (Experiment Setup):
>
> **Hyper-parameters.** We use grid search to optimize hyperparameters, including learning rate, batch size, step size of MoDo, and penalty parameter increasing rate. For both SSL pre-training and supervised fine-tuning, the backbone learning rate is consistently set higher than the classification parameter learning rate. The SSL pre-training phase starts with a learning rate of $\alpha = 5 \times 10^{-4}$ for 100 epochs, annealed by a factor of 0.1 every 20 epochs. Fine-tuning uses a maximum learning rate of $\beta = 5 \times 10^{-5}$, with a scheduler reducing the learning rate by a factor of 0.1 if the test loss does not improve within 10 epochs. All multi-objective models (VS-MSP, VC-MSP, and VM-MSP) and joint PT+FT models are trained for 200 epochs. For PT+FT, we pre-train the model for 200 epochs and fine-tune it for an additional 100 epochs. A batch size of 256 and AdamW optimizer are used for both self-supervised and supervised training. The same hyperparameter settings are applied across all training methods to ensure consistency and comparability.
>
>
> **Q4: The appendix may clear up confusion; main body should be clearer.**
>
> We moved essential definitions and mechanisms to the main body: (i) conflict‑avoiding direction and its MoDo realization, (ii) the constrained formulation and its penalty‑based counterpart, and (iii) explicit per‑method backbone updates (Table 1) so readers need not consult the appendix to follow the training procedure. Appendix B provides derivations; the main text already contains the practical rules and schedules.
>
>
> **Q5: Novelty is limited but the empirical evidence is clear.**
>
> Thank you for the assessment of the empirical gains. While our system leverages familiar components (shared encoder, task‑specific heads), our contribution is **not** a re‑branding of curriculum or static weighting. What is novel is how we **structure, operationalize, and measure** conflict in multilingual ASR–ST using a multilevel multi‑objective lens:
>
> 1. **Formulation‑level novelty.**
>    We cast multilingual ASR–ST as a **multilevel (lexicographic)** MOO problem in which the self‑supervised objective serves as an **ε‑constraint**, and supervised tasks are **separated across levels**. Unlike single‑level multi‑tasking or curricula with time‑varying weights, our **penalty‑based realization** yields **single‑loop** gradient updates that *implement* the hierarchy without nested solves (see Table 1).
>
> 2. **Actionable hierarchy selection & diagnostics.**
>    We show that **which objective resides at which level matters**, and we provide a **data‑driven rule**: measure gradient conflict, place objectives accordingly, and **anneal penalties** to encode priorities. Ablations justify the chosen hierarchy and schedule (Sec. 4).
>
> 3. **Efficiency via targeted conflict handling.**
>    We introduce **layer‑wise conflict selection** that computes dynamic weights **only where conflict is observed**, reducing compute/memory overhead **without** degrading WER/BLEU (Sec. 4.7).
>
> 4. **Systematic conflict analysis for multilingual ASR–ST.**
>    We provide a **comprehensive characterization** of conflict across languages, tasks, and layers, explaining *why* multilevel separation helps here and when it should transfer to other architectures (Sec. 4; Limitations).
>
> Together, these elements form a principled, practically deployable recipe: (i) measure conflict, (ii) place objectives across levels accordingly, (iii) enforce the lower‑level via a penalty schedule, and (iv) restrict dynamic weighting to conflicting layers. We believe this constitutes a substantive methodological contribution beyond conventional task‑specific heads or generic curricula.
>
> **Q6: Gradient conflicts don’t change much during training.**
>
> Great observation! We observe a similar phenomenon under two‑stage training (without multi-objective optimization): conflicts persist in the same layers (Figure 3). Our method addresses this by (i) separating highly conflicting objectives across levels (VC/VM‑MSP), and (ii) targeting the conflicting layers when computing the update direction, which is precisely where conflicts concentrate.

---

> > ### Comment · Reviewer_UXqi · 2025-08-04
> >
> > Thank you for clarifying some of the points and your efforts to make the paper clearer.

---

> > > ### Author Response · Authors · 2025-08-08
> > >
> > > Thank you for your kind words and for recognizing our efforts to improve the clarity of the paper. We appreciate your feedback and remain open to further suggestions to make it even stronger.

---

### Official Review · Reviewer_Ljor · 2025-06-26

**Clarity:** 2
**Significance:** 2
**Originality:** 2
**Rating:** 4
**Confidence:** 4

**Summary:**

This paper focuses on the training of multilingual speech processing models that perform both transcription and translation for multiple languages. The challenge is that each task has its own objective function, and therefore this can be viewed as a multi-objective optimization problem. The standard approach is to either (1) do multistep pre-training and fine-tuning phases with different objectives on different data or (2) linearly combine all objectives into a single objective. The authors argue that a more nuanced multi-objective approach is better.

They propose three training methods variants (VS-MSP, VC-MSP, VM-MSP) and perform experiments on CoVOST2 and LibriSpeech/AIShell. Results in WER and BLEU are generally promising.

**Questions:**

**W1.** The models used here have a large set of what authors called backbone parameters \theta that are shared across language/tasks and another collection of language/task-dependent parameters \phi. The \theta parameter is involved in the unsupervised loss term, whereas \phi are involved in the various supervised loss terms. I believe the fact that there are independent \phi parameters for different objective means that there is not so much "gradient conflict" in the objectives that are most likely to have conflict.

For example, suppose I have a single decoder that performs both translation and transcription, then it is likely the two different objectives will cause conflict. However, in the conformer model for example, the translation is performed by the decoder while the transcription is performed with CTC on the encoder. Another example: if transcription for Chinese and English is part of the same CTC with a shared vocabulary, then there is higher chance of gradient conflict. However, if the two languages are handled by different heads, then there will be less conflict.

In other words, I think the proposed methods and results are tied to a certain type of architecture that has a nice division of shared and task-dependent parameters. If the authors agree with this assessment, I would recommend the following changes to the paper:

 - Discuss the model architecture in more detail (and include Figure 7) in the main text
 - Add the disclaimer that the proposed methods are most suited for this kind of architecture. Add in the Limitation how things might differ for different architectures.


**W2.** The appendix is quite long. I usually do not need to pay too much attention to the appendix in reviewing but I find that for this paper I needed to read the appendix in detail to understand what is being done. Below are some parts that I think deserve to be in the main text:
 - Technical discussion of static vs dynamic weighting
 - How Equations 5-8 actually optimizes the objectives that are defined (especially it is hard to see how the constraints are handled just from the main text)
 - Figure 5 motivating some amount of gradient conflict in practice.
 - Figure 7 and details of the model architecture as mentioned previously

To reclaim space, I think the following parts of the main text can be reduced:
 - I don't think Fig 1 actually reflects what is happening in practice and can be removed. For example, do we really know that the unsupervised objective has broader support like shown in the green curve? If not, it is best not to include it.
 - There is significant repetition in Sec 3 and Sec 4 first trying to motivate methods in the abstract then adapting it to the specific setting of speech processing. I think it is best just to explain everything in the specific setting considering W1.

**W3.** I think the language needs to be rigorous when talking about optimization. Here are some examples that bothered me a bit:
 - Line 565 "ensuring guaranteed convergence to Pareto stationary solutions"; Line 156 "ensuring the attainment of a Pareto stationary point." -> Is there really guarantees in convergence or attainment? If not, better to say "encouraging convergence"
 - Line 120: "while uniquely narrowing the search region of the optimal solutions to a representation preserving subspace." First what is a representation preserving subspace? Second, does your update rule actually narrow the search region?
 - Line 148: "A higher degree of conflict would undermine its role in narrowing the search space for a common optimal point". I think this doesn't fit with the definition of pareto optimality, which by construction is about defining what counts as "optimal" despite conflicts in objectives.
 - Eq 4. has multiple nested "such that" constraints. It's understandable what is meant, but is there a better way to write this?

**W4.** Currently the related work sections in the main text and appendix covers "Multilingual speech recognition and translation" and "Multi-task Learning for Speech Recognition" nicely, but I would also recommend adding a paragraph on "Multi-objective Optimization".  First, there have been some work that explored multi-objective optimization for speech processing which may be worth mentioning. For example, [1] performs multi-objective optimization of hyperparameters on accuracy and size/speed objectives. The actual method is very different from yours, but it may be worth mentioning how the problem differs. Second, your proposed method may be viewed as a form of multi-objective optimization called lexicographic method [2]. So it will be good to cite that. Another related citation is [3] which is also similarly motivated, though it is focused on RL.

 - [1] Evolution-Strategy-Based Automation of System Development for High-Performance Speech Recognition https://ieeexplore.ieee.org/abstract/document/8470178
 - [2] Survey of multi-objective optimization methods for engineering https://link.springer.com/article/10.1007/s00158-003-0368-6
 - [3] Lexicographic Multi-Objective Reinforcement Learning https://arxiv.org/pdf/2212.13769

**Ethical Concerns:**

["NO or VERY MINOR ethics concerns only"]

**Final Justification:**

I've raised the score from 3 to 4 reflecting author's satisfactory response.

**Limitations:**

Currently there is not much discussion of Limitation (besides " further theoretical analysis would be an
interesting direction for future research".) As mentioned in W1, talking about the results and methods being possibly dependent on the architecture would be worth mentioning as a Limitation (assuming the authors agree).

**Quality:**

3

**Strengths And Weaknesses:**

Strengths:
 - S1. Tackles a well-known challenge in multilingual speech processing
 - S2. Performs a comprehensive set of experiments using different Conformer and Whisper models

Weakness:
 - W1. The positive results are probably tied to the architecture used. I am not sure if the VS-MSP, VC-MSP, VM-MSP methods make sense if the architecture does not assume a backbone and different heads for different objectives.
 - W2. Many of the important and interesting details are in the appendix. The main text itself was not as insightful as it could be.
 - W3. Some of the description/language relating to optimization is not rigorous.
 - W4. Related work in multi-objective optimization can be made more detailed.

All these are points that can be addressed with a writing revision. Please see below for suggestions.

---

> ### Author Rebuttal · Authors · 2025-07-29
>
> We thank the reviewer for recognizing the strengths of our paper, which include addressing a well-known challenge in multilingual speech processing. We also appreciate the detailed review that will further strengthen our work. We hope that our responses to your comments below fully address your concerns.
>
> **Q1: Results are tied to a certain type of architecture.**
>
> We appreciate the reviewer’s thoughtful question. In our study, we evaluated the proposed method across three speech models—**Conformer**, **Wav2Vec2**, and **Whisper**—all of which are widely employed in speech processing [1,2].
>
> While the use of separate heads may reduce gradient conflict at the output level, it does **not eliminate conflicts within the shared encoder**, which contains the majority of the model parameters. As shown in **Figure 3**, we observe significant gradient conflicts even when task-specific heads are used, highlighting that the shared representation learning remains a critical bottleneck. Our multi-objective optimization algorithm is specifically designed to **address these conflicts in the shared layers**.
>
> We agree that our current results and analysis are grounded in architectures with a modular separation between shared and task-specific parameters. However, we note that this architectural pattern is **prevalent in modern multilingual and multitask speech systems**. Investigating how our method behaves under alternative designs—such as models with a **shared decoder** for both transcription and translation—is beyond the scope of this paper, and is a valuable direction for future work.
>
> Following the reviewer’s suggestion, we have added the following into the **Conclusions and Limitations** section:
>
> >**Our evaluation focused on architectures that exhibit a clear separation between shared and task-specific parameters—for example, a shared encoder followed by distinct classification heads for ASR and translation. This architectural pattern is common in speech processing, but how the method generalizes to more tightly coupled architectures—such as models with a unified decoder shared across tasks—remains an open question. In such cases, task-specific conflicts may propagate more deeply through shared components, potentially requiring alternative optimization strategies or additional regularization. Investigating these scenarios is a promising direction for future research.**
>
> > 1. Burchi, M. et al. "Multilingual audio-visual speech recognition with hybrid ctc/rnn-t fast conformer."
> > 2. De Zuazo, X. et al. "Whisper-LM: Improving ASR Models with Language Models for Low-Resource Languages."
>
>
> **Q2: The appendix is quite long.**
>
> Following the reviewer’s suggestion, we have moved the following parts into the main text.
>
> **i) Static vs. dynamic weighting:** We moved the discussion of static vs. dynamic weighting from Appendix B to Section 3.
>
> **ii) Equations 5-8:** We moved the explanation of Equations 5–8 from the appendix to Section 3 for clarity.
>
> Specifically, we clarify that Equations 5–8 describe a **penalty-based multilevel optimization**, where each objective is optimized sequentially while incorporating lower-priority objectives as penalties, scaled by a time-varying parameter $\eta$. This mechanism encourages the model to prioritize upper-level objectives while still accounting for progress on lower levels.
>
> **iii) Fig 1** is intended as a **conceptual illustration** adapted from Fig 3.2.1 in [1]. In that figure, the geometry of constrained multi-objective optimization is used to show how varying the constraint bound affects the feasible set of the primary objective.
>
> In our adaptation, we use the CPC loss as a constraint-like term that influences the optimization landscape of the main tasks. The figure does **not represent actual empirical distributions**, but rather **visualizes the intuition** that if the unsupervised constraint is too tight, it may overly restrict the optimization; whereas if it is too loose, it may have negligible influence. As this figure is a conceptual illustration, we have removed the figure and directly cited [1].
>
> **iv) Fig 5 and 7:** We moved these figures to Section 5 (Experimental Results and Findings) and added corresponding descriptions.
>
> **v) Sec 3 and 4:** In the revised manuscript, we have **merged Sec 3 and 4 into a single unified section** to streamline the presentation and eliminate redundancy. Repeated explanations and motivating sentences have been removed or consolidated for clarity.
>
> Hope our revision will address your concern on the writing.
>
> **Q3: Language needs to be rigorous.**
>
> To make the language more rigorous we have made the following changes:
>
> **Line 565:** We have replaced "ensuring the attainment..." with "encouraging convergence".
>
> **Line 120:** We appreciate the clarification request. We have (i) replaced **“subspace”** with **“representation‑preserving neighborhood”** and (ii) clarified how “narrowing” is implemented.
>
> - **“Representation‑preserving neighborhood.”**
>   We define it as the **sublevel set** of the self‑supervised loss around its minimum:
>
> $$\mathcal{R}_\delta := \\{\theta: \ell_u(\theta) \le \ell_u^\star + \delta\\}, \quad \ell^*_u := \min\_{\theta} \ell_u(\theta).$$
>
> Staying in $\mathcal{R}_\delta$ keeps the encoder representation close to the invariances encouraged by the self‑supervised objective.
>
> - **Does the update actually “narrow” the search region?**
>   We do **not** claim a projection onto $\mathcal{R}_\delta$. Instead, our training uses a **penalized surrogate** of the $\varepsilon$‑constraint method ([1], 3.2): we add the penalty term $\eta\left(\ell_u(\theta)-\ell_u^\star\right)$ to the vector objective. Increasing $\eta$ **restricts** the iterates to remain near $\mathcal{R}\_\delta$. In the limit of large $\eta$ (or with an explicit $\varepsilon$‑constraint), this corresponds to solving the $\varepsilon$‑constraint formulation, which **does** reduce the feasible set. Our revised text therefore states:
>   **“restricting the search region to a representation‑preserving neighborhood that continues to intersect the Pareto set,”** emphasizing a restriction rather than a literal subspace or a guaranteed projection.
>
> > 1. Miettinen, K. "Nonlinear multiobjective optimization." 1999.
>
> **Line 148:** We agree that Pareto optimality is defined independently of the degree of conflict among objectives. Our statement is not about the existence of the Pareto set but about the algorithmic effect of gradient conflict on gradient‑based multi‑objective optimization. In methods that seek a common descent direction (e.g., MGDA\MoDo‑type updates), the set of feasible common descent directions
> $$\mathcal{C}(\Theta):=\\{d:\left<\nabla\ell_m(\Theta),d\right>\leq0, \forall m\\}$$
>
> **shrinks as gradient conflict increases** (i.e., as pairwise cosine similarities become more negative). When this cone is small or degenerate, it becomes harder to make simultaneous progress on all objectives, even though Pareto‑optimal solutions still exist.
>
> Our lower‑level constraint is introduced precisely because—empirically—it is mostly aligned with the supervised objectives (Appendix E, Fig. 5). Under such alignment, constraining it guides the optimization to a “representation‑preserving” region that still intersects the Pareto set while increasing the chance that a computable common descent direction exists. If, counterfactually, the lower‑level objective was strongly conflicting, then constraining it would indeed be unhelpful for the solver—this is exactly the point we intended to convey.
>
> To avoid any misunderstanding, we have revised the manuscript to make clear that our claim concerns optimization dynamics and the size of the common‑descent cone, not the definition or existence of Pareto‑optimal solutions.
>
> >**its gradient update direction should exhibit minimal conflict with the upper‑level objectives. When such alignment holds, the constraint restricts the feasible region to a representation‑preserving neighborhood that still intersects the Pareto set and admits a common descent direction for gradient‑based solvers; if the lower‑level objective were strongly conflicting, the constraint would not aid optimization.**
>
> **Eq 4:** Thank you for the suggestion. We agree that alternative displays can be clearer. For consistency with the rest of our notation and to avoid ripple changes across sections, we keep Eq. (4) as written in this revision. As the reviewer notes, the current form is understandable.
>
> **Q4: Related work section.**
>
> We have added the following in the related work section:
>
> **Multi-objective optimization** aims to optimize multiple, often conflicting, objectives simultaneously and has been applied in various domains including engineering [1], finance [2], and decision making [3]. Multi-objective optimization has also been used for solving
> lexicographic multi-objective problems [4,5]. It has also seen applications in speech processing. In previous work, multi-objective optimization has been applied at the system development level, using evolutionary strategies to jointly optimize recognition accuracy and model size by tuning meta-parameters such as model topology and training configuration [6]. In contrast, our work focuses on loss-level conflicts during the training of multilingual and multitask models, where multiple learning signals interact within a shared backbone.
>
> > 1. Cheng, Y. "Multiobjective optimum design of structures with genetic algorithm and game theory: Application to life-cycle cost design."
> > 2. Doumpos, M. et al. "Multi-objective optimization models in finance and investments."
> > 3. Deb, K. et al. "Multi-objective optimization."
> > 4. Marler, R. el al. "Survey of multi-objective optimization methods for engineering."
> > 5. Skalse, J. et al. "Lexicographic multi-objective reinforcement learning."
> > 6. Moriya, T. et al. "Evolution-strategy-based automation of system development for high-performance speech recognition."

---

> > ### Comment · Reviewer_Ljor · 2025-08-05
> >
> > Thanks for the response. I think the paper will look better with these addressed. I've raised the score from 3 to 4.

---

> > > ### Author Response · Authors · 2025-08-08
> > >
> > > Thank you for your thoughtful feedback and for raising the score. We appreciate your constructive suggestions and will work on addressing them to further improve the paper. We’re happy to continue the discussion if you have any additional comments.

---

### Official Review · Reviewer_95Wr · 2025-06-29

**Clarity:** 2
**Significance:** 3
**Originality:** 3
**Rating:** 4
**Confidence:** 4

**Summary:**

In this work, the authors focus on the training interference among self-supervised training, speech recognition with CTC and speech translation with cross entropy criteria. They propose to consider the multi-objection optimization problem as multi-level multi-task optimization problem and penalize the minor level tasks, such as unsupervised and cross entropy for the translation task. The proposed training strategy shows good improvement for both multilingual  ASR and ST tasks at the cost of more memory and training time. A simple and light weight layer selection strategy is also examined and reduce the training memory and time significantly. However, there are many details are missing in the main paper, which are included in the appendix. Also, the experiment should include baselines from other research work for a comprehensive comparison.

**Questions:**

- two optimization sequences: USA and UAS, are discussed in the Table 3 and 4. They are not explained properly. Do ``USA'' mean the model is optimized according equ. 8 (Translation as minor task) while ``UAS'' chooses the ASR CTC task as secondary task and optimized with penalty parameter?
- Do you try to use cross entropy to optimize the ASR task too? It may increase the backbone model shared between ASR and ST.

Typo:
line 240: "VC-MSP, the bilevel penalty is initialized to η0 = 0 and increased by 0.02 each epoch" -> "VC-MSP, the bilevel penalty is initialized to η = 0 and increased by 0.02 each epoch"

**Ethical Concerns:**

["NO or VERY MINOR ethics concerns only"]

**Quality:**

3

**Strengths And Weaknesses:**

Strengths:
- the multi-objection optimization problem is considered as a multi-level multi-task optimization problem with penalization applied to minor level tasks.
- the proposed method achieves good improvement compared with the baselines
- an efficient method is proposed to reduce the computation cost during training.

Weaknesses:
- There are many details are missing in the main paper, such as the efficient method MSP method. Thought those information is included in the appendix.
-  The experiment should include baselines from other research work for a comprehensive comparison.

---

> ### Author Rebuttal · Authors · 2025-07-31
>
> We thank the reviewer for recognizing the strengths of our paper, which include achieving significant improvements and efficiently reducing computational cost. We hope our responses to your comments below address your concerns.
>
>
> **Q1: Details are missing in the main paper.**
>
> In the revised manuscript, we have incorporated key content from the appendix into the main body of the paper to improve clarity and self-containment. Specifically:
>
> 1. The technical discussion of **static vs. dynamic weighting** has been moved from Appendix B into **Section 3** of the main text.
> 2. We have **merged Sections 3 and 4** to improve narrative flow and better integrate the core optimization formulation with its motivation.
> 3. Full descriptions of the **Conformer** and **Whisper** models have been moved from Appendix G into **Section 5**, ensuring architectural details are accessible.
> 4. **Figure 5** has been relocated from Appendix E into **Section 5.1**, where it now serves as a key component in motivating our method.
>
> These changes aim to make the paper easier to follow and ensure that all essential methodological and experimental components are included in the main text.
>
> **Q2: The experiments should include baselines from other research work for a comprehensive comparison.**
>
> We agree that comprehensive baselines are essential for evaluating generalization. In addition to using the Conformer model, our experiments include two **widely recognized and competitive models** from prior work: **Wav2Vec2** and **Whisper**. These models are strong baselines with proven effectiveness in multilingual and multi-task speech processing tasks [1,2,3]. To ensure fairness, we evaluated all models under the **same experimental conditions**, including training data and optimization setup.
>
> In future work, we plan to extend our study to larger datasets, more tasks, and additional model architectures to further evaluate the generality of our approach.
>
> > 1. Radford, A. et al. "Robust speech recognition via large-scale weak supervision."
> > 2. Pratap, V. et al. "Scaling speech technology to 1,000+ languages."
> > 3. Team, Gemini, et al. "Gemini: a family of highly capable multimodal models."
>
> **Q3: two optimization sequences: USA and UAS, are discussed in Tables 3 and 4. They are not explained properly.**
>
> We thank the reviewer for this observation.  The descriptions of **USA** and **UAS** are originally provided in **Section 5.2, line 267**, and follow the optimization ordering principle used in **penalty-based optimization**. Specifically:
>
> - **UAS** refers to the optimization sequence: *Unsupervised → ASR → Translation*. That is, we first optimize the self-supervised loss, then the ASR objective (CTC loss), and finally the translation objective (cross-entropy loss).
> - **USA** refers to the sequence: *Unsupervised → Translation → ASR*. That is, we first optimize the self-supervised loss, then the translation objective, and finally the ASR objective.
>
> These sequences reflect different priority orderings in the multilevel structure (Equation 8), where upper-level objectives are weighted more heavily, and lower-level objectives are included as penalty terms.
>
> To avoid confusion, we have **explicitly added** the following clarification to the updated version of the paper (in Section 5.2):
>
> > “We tested two optimization sequences: UAS (self-supervised → speech recognition → translation) and USA (self-supervised → translation → recognition). UAS means the unsupervised loss is optimized first, followed by the ASR objective (CTC loss), and then the translation objective (cross-entropy loss). For USA, the unsupervised objective is optimized first, followed by the translation objective and then ASR.”
>
> We hope this resolves the concern and makes the optimization orderings clearer.
>
> **Q4: Do you try to use cross entropy to optimize the ASR task too? It may increase the backbone model shared between ASR and ST.**
>
> We appreciate the reviewer’s suggestion. In our experiments, we intentionally use **CTC** as the objective for ASR, rather than cross-entropy. CTC is a **sequence alignment–based criterion** that is specifically designed to handle input-output length mismatches without requiring frame-level alignment or teacher forcing, making it particularly well-suited for ASR [1,2,3].
>
> While using cross-entropy for ASR could lead to tighter coupling between ASR and ST objectives by sharing a decoder, it would also require architectural changes (e.g., adding an autoregressive decoder for ASR). Therefore, we follow standard practice by using **CTC for ASR** and **cross-entropy for ST**, which also aligns well with our goal of analyzing and mitigating gradient conflicts between distinct objectives.
>
> > 1. Watanabe, S. et al. "Hybrid CTC/attention architecture for end-to-end speech recognition."
> > 2. Baevski, A. et al. "wav2vec 2.0: A framework for self-supervised learning of speech representations."
> > 3. Gulati, A. et al. "Conformer: Convolution-augmented Transformer for Speech Recognition."
>
> **Q5: Typo: line 240**
>
> Thanks a lot for catching the typos! We have corrected the typo.

---

### Official Review · Reviewer_mmrV · 2025-07-06

**Clarity:** 3
**Significance:** 3
**Originality:** 3
**Rating:** 5
**Confidence:** 4

**Summary:**

Multi-language and Multi-task speech processing remains as a very challenging issue. In this paper, author cope with this challenge via Multi-objective optimization.  Typically in the literature researchers tend to use one loss term for each task, for instance, a prediction loss to learn a good representation, language-specific CTC for transcription, cross-entropy for translation and fairness constraints for underrepresented languages.   Multi-objective optimization  provides a principled approach for handling conflicting training objectives. More specifically, this paper proposes to first optimize the lower level loss , the self-supervised loss l_u. Second, it feed its solution into upper level losses such as combining CTC and cross entropy. This hierarchical approach can keep moving on . Experimental results show its effectiveness.

**Questions:**

Is this approach applicable to broad speech tasks such as speaker verfication and speech enhancement? How reboust is it?

**Ethical Concerns:**

["NO or VERY MINOR ethics concerns only"]

**Limitations:**

Yes

**Paper Formatting Concerns:**

fold--> feed  and other typos

**Quality:**

3

**Strengths And Weaknesses:**

Strengths:
Proposed a workable solution to hand the key problem facing speech processing.

Weaknesses:
The proposed solution looks fragile in real scenarios since it require carefully designing the hierarchical combination.

---

> ### Author Rebuttal · Authors · 2025-07-31
>
> We thank the reviewer for recognizing the strengths of our paper and noting that the proposed solutions can effectively address the key problem in speech processing. We hope our responses below to your comments address your concerns.
>
> **Q1: The proposed solution looks fragile in real scenarios since it requires careful design of the hierarchical structure.**
>
> We propose and investigate a principled and practically deployable recipe: (i) measure conflict, (ii) place objectives across levels accordingly, (iii) enforce the lower‑level via a penalty schedule, and (iv) restrict dynamic weighting to conflicting layers.
>
> Our core empirical finding is that **the order** in which conflicting objectives are optimized—the **hierarchy**—influences model performance in multilingual multi‑task speech processing. Rather than relying on ad-hoc heuristics, we use **gradient cosine similarity** (Figure 5) to identify strongly conflicting objective pairs (e.g., ASR CTC vs. translation CE). This conflict analysis directly informs our grouping of objectives into separate optimization levels.
>
> Leveraging this insight, we develop algorithms through a **multilevel penalty‑based formulation**, where a time‑varying penalty parameter $\eta$ dynamically controls the influence of lower‑level objectives. As shown in **Table 5** of the paper, tuning $\eta$ via standard hyperparameter‑search procedures yields stable progress across levels **without manual intervention**. This dynamic control makes our scheme robust across penalty schedules.
>
> We have extensively evaluated this hierarchical structuring of conflict objectives under a wide variety of model architectures, sizes, languages, and penalty schedules for multilingual speech recognition and translation, and observed consistent performance improvements. Therefore, we believe this would be a robust recipe in practice.
>
> **Q2: Is this approach applicable to broad speech tasks such as speaker verification and speech enhancement? How robust is it?**
>
> Yes, our algorithms extend naturally to additional speech tasks with minimal modifications. Once a new task is defined by its corresponding loss with the task-specific dataset, it can be incorporated into the existing multilevel structure without altering the core optimization algorithms. The shared parameters will continue to be updated using the same conflict-aware optimization procedure already described. This flexibility reflects one of the strengths of our algorithm—it scales to more tasks without requiring architectural changes or task-specific adaptation.
>
> For example, to add the following tasks:
>
> - **Speaker Verification**
>   Add a cross-entropy loss: $\ell_{\text{sv}}(\theta, \phi_{\text{sv}})$
>   where $\phi_{\text{sv}}$ are the speaker-verification head parameters.
> - **Speech Enhancement**
>   Add a mean-squared error loss: $\ell_{\text{se}}(\theta, \phi_{\text{se}})$
>   where $\phi_{\text{se}}$ are the enhancement-decoder parameters.
>
> The overall shared-parameter update (cf. Eq. 8) incorporates these new losses:
>
> $$\theta^{k+1} = \theta^k - \alpha \Bigl( L_{\mathrm{ASR}} + L_{\mathrm{ST}} + L_{\mathrm{SVE}} + L_{\mathrm{Unsupervised}} \Bigr),$$
> where
> $$L_{\mathrm{ASR}} = \sum_{t=1}^{T} \lambda^k_{t,\mathrm{ctc}} \nabla_\theta \ell_{\mathrm{ctc}},$$
> $$L_{\mathrm{ST}} = \eta_1 \sum_{t=1}^{T} \lambda^k_{t,\mathrm{ce}} \nabla_\theta \ell_{\mathrm{ce}},$$
> $$L_{\mathrm{SVE}} = \eta_2 \bigl( \lambda^k_{\mathrm{sv}} \nabla_\theta \ell_{\mathrm{sv}} + \lambda^k_{\mathrm{se}} \nabla_\theta \ell_{\mathrm{se}} \bigr),$$
> $$L_{\mathrm{Unsupervised}} = \eta \nabla_\theta \ell_{\mathrm{u}}.$$
>
> Our simulations confirm that with the existing CTC, CE, and CPC objectives for ASR and ST tasks, our algorithm successfully mitigates conflicts among these tasks—demonstrating its robustness and broad applicability.
>
> To further evaluate its extensibility to broader speech tasks, we conducted an additional experiment by adding a gender-classification task alongside multilingual ASR, using both the LibriSpeech train-clean-100 and AISHELL datasets. For gender classification, we trained our model on the LibriSpeech train-clean-100 split.
>
> Our results demonstrate that our proposed algorithms—VM-ASR variants **UECG** and **UCEG**—consistently outperform baseline methods across both ASR and gender classification tasks. In the **UECG** variant, we first optimize the unsupervised loss, followed by the English ASR loss, then the Chinese ASR loss, and finally the gender classification loss. In contrast, the **UCEG** variant follows a different optimization order: we first optimize the unsupervised loss, then the Chinese ASR loss, followed by the English ASR loss, and finally the gender classification loss.
>
>
> | Param | Task | Lang | Two-stage (PT+FT) | Joint PT+FT W/O MOO | VS-ASR | VC-ASR | VM-ASR (UECG) | VM-ASR (UCEG) |
> |-------|------|------|-------------------|--------------------|--------|--------|--------------|--------------|
> | **58M** | **ASR (WER)** | En (test-clean) | 8.0% | 7.3% | 7.7% | 7.2% | 6.9% | **6.8%** |
> |         |                 | En (test-other) | 18.3% | 17.8% | 18.1% | 17.6% | 17.5% | **17.3%** |
> |         |        **ASR (CER)**         | Zh              | 7.9%  | 7.4%  | 7.6%  | 7.1% | **6.6%** | 6.7% |
> |         |                 | **Average**     | 11.4% | 10.8% | 11.1% | 10.6% | 10.3% | **10.2%** |
> | **58M** | **Gender Classification (Accuracy)** | En (test-clean) | 97.3% | 97.4% | 97.6% | 97.7% | 98.1% | **98.2%** |
> |         |                 | En (test-other) | 90.8% | 91.4% | 92.3% | 92.8% | 93.2% | **93.4%** |
>
> **Q3: fold--> feed and other typos**
>
> Good catch! We have corrected the typos in the latest version of the paper.

---

### Decision · Program_Chairs · 2025-09-17

**Decision:**

Accept (poster)

**Comment:**

**Summary**

This paper identifies the challenges of optimizing for multiple speech tasks simultaneously.  These tasks have distinct objective functions and the authors explore different approaches to combining these.

**Reasons to accept**

The reviewers were consistently positive on this work. They identified this as an important task, and the proposed analyses and solutions to be well motivated.  Ljor summarized this as follows “the contribution are not a new exciting concepts, but the experiments are sound and can serve as good reference for people working on this problem.”.

**Reasons not to accept**

General lack of enthusiasm, and a concern that the findings of this work won’t scale sufficiently well to influence other work. This leads to limitations of both novelty and significance.

**Decision rationale**

The reviewers are quite positive on this work.  The limitations of the work stem primarily from the scope of the problem that’s being addressed.  There is good analysis and well motivated experiments that have been brought to bear on a particular problem.  These are sufficient enough reasons to accept.  To be more convinced in this decision, I would look for a more obvious statement of significance on this work addressing the question “why will a future reader cite this work?”  While I believe there are answers to this question, they are not centered by the authors themselves sufficiently to make it obvious to the reviewers or to me.